# Topological network analysis of patient similarity for precision management of acute blood pressure in spinal cord injury

**Abel Torres-Espín[1†], Jenny Haefeli[1†], Reza Ehsanian[2], Dolores Torres[1], Carlos A Almeida[1], J Russell Huie[1,3], Austin Chou[1], Dmitriy Morozov[4], Nicole Sanderson[5], Benjamin Dirlikov[6], Catherine G Suen[1], Jessica L Nielson[7,8], Nikos Kyritsis[1], Debra D Hemmerle[1], Jason F Talbott[9], Geoffrey T Manley[1], Sanjay S Dhall[1], William D Whetstone[10], Jacqueline C Bresnahan[1,3], Michael S Beattie[1,3], Stephen L McKenna[11,12], Jonathan Z Pan[13]\*, Adam R Ferguson[1,3]\*, The TRACK-SCI Investigators[1]**

[1]Weill Institute for Neurosciences; Brain and Spinal Injury Center (BASIC), Department of Neurological Surgery, University of California, San Francisco; Zuckerberg San Francisco General Hospital and Trauma Center, San Francisco, United States; [2]Division of Physical Medicine and Rehabilitation, Department of Orthopaedics and Rehabilitation, University of New Mexico School of Medicine, Albuquerque, United States; [3]San Francisco Veterans Affairs Healthcare System, San Francisco, United States; [4]Computational Research Division, Lawrence Berkeley National Laboratory, Berkeley, United States; [5]Lawrence Berkeley National Laboratory, Berkeley, United States; [6]Rehabilitation Research Center, Santa Clara Valley Medical Center, San Jose, United States; [7]Department of Psychiatry and Behavioral Science, and University of Minnesota, Minneapolis, United States; [8]Institute for Health Informatics, University of Minnesota, Minneapolis, United States; [9]Department of Radiology and Biomedical Imaging, University of California, San Francisco, San Francisco, United States; [10]Department of Emergency Medicine, University of California, San Francisco; Zuckerberg San Francisco General Hospital and Trauma Center, San Francisco, United States; [11]Department of Physical Medicine and Rehabilitation, Santa Clara Valley Medical Center, San Jose, United States; [12]Department of Neurosurgery, Stanford University, Stanford, United States; [13]Department of Anesthesia and Perioperative Care, University of California, San Francisco; Zuckerberg San Francisco General Hospital and Trauma Center, San Francisco, United States

**\*For correspondence:**
jonathan.pan@ucsf.edu (JZP);
adam.ferguson@ucsf.edu (ARF)

†These authors contributed equally to this work

**Group author details:**
The TRACK-SCI Investigators See page 22

## Abstract

**Background:** Predicting neurological recovery after spinal cord injury (SCI) is challenging. Using topological data analysis, we have previously shown that mean arterial pressure (MAP) during SCI surgery predicts long-term functional recovery in rodent models, motivating the present multicenter study in patients.

**Methods:** Intra-operative monitoring records and neurological outcome data were extracted (n = 118 patients). We built a similarity network of patients from a low-dimensional space embedded using a non-linear algorithm, Isomap, and ensured topological extraction using persistent homology metrics. Confirmatory analysis was conducted through regression methods.

**Results:** Network analysis suggested that time outside of an optimum MAP range (hypotension or hypertension) during surgery was associated with lower likelihood of neurological recovery at

hospital discharge. Logistic and LASSO (least absolute shrinkage and selection operator) regression confirmed these findings, revealing an optimal MAP range of 76–[104-117] mmHg associated with neurological recovery.

**Conclusions:** We show that deviation from this optimal MAP range during SCI surgery predicts lower probability of neurological recovery and suggest new targets for therapeutic intervention.

**Funding:** NIH/NINDS: R01NS088475 (ARF); R01NS122888 (ARF); UH3NS106899 (ARF); Department of Veterans Affairs: 1I01RX002245 (ARF), I01RX002787 (ARF); Wings for Life Foundation (ATE, ARF); Craig H. Neilsen Foundation (ARF); and DOD: SC150198 (MSB); SC190233 (MSB); DOE: DE-AC02-05CH11231 (DM).

## Editor's evaluation

The major strengths of this paper are the use of a combination of relatively novel approaches/applications to the identification of important predictors for recovery after spinal cord surgery. These include various data reduction techniques such as dissimilarity matrices and a subject-centered topological network analysis to identify phenotypes.

## Introduction

Spinal cord injury (SCI) may result in motor, sensory, and autonomic dysfunction in various degrees, depending on the injury severity and location. In the USA, the incidence of SCI is around 18,000 new cases per year, with a total prevalence ranging from 250,000 to 368,000 cases (*National Spinal Cor Injury Statistical Center, 2021*). The dramatic life event of SCI imposes a high socioeconomic burden, with an estimated lifetime cost between $1.2 and $5.1 million per patient (*National Spinal Cor Injury Statistical Center, 2021*).

Prior retrospective observational single-center studies in humans suggest that lower post-surgery mean arterial pressure (MAP) predicts poor outcome (*Cohn et al., 2010*; *Hawryluk et al., 2015*; *Saadeh et al., 2017*; *Ehsanian, 2020*), which have resulted in clinical guidelines focused on avoidance of hypotension by maintaining MAP >85 mmHg during the first 7 days of injury (*Aarabi et al., 2013*). The rational for MAP augmentation to avoid hypotension is based on the hypothesis that decreased spinal cord prefusion leads to ischemia and additional tissue lost (*Mautes et al., 2000*; *Soubeyrand et al., 2014*). Importantly, experimentally raising MAP during acute SCI in animals by using vasopressors increases the risk for hemorrhage and consequent tissue damage (*Soubeyrand et al., 2014*; *Streijger et al., 2018*; *Guha et al., 1987*). In acute cervical patients with SCI, spinal cord hemorrhage correlates with poor prognosis for neurological recovery (*Miyanji et al., 2007*). These findings collectively suggest that hypo- and hypertension must be accounted for in MAP management, but there remains a gap in evidence from clinical studies that definitively informs MAP guidelines (*Saadeh et al., 2017*).

One of the challenges resulting in the lack of strong evidence for MAP management in patients with acute SCI is the heterogeneity of injury. The heterogeneity of SCIs results in data complexity that benefit from modern analytic tools. Using the machine intelligence approach of topological data analysis, we have previously shown that hypertension during SCI surgery (ultra-acute phase) predicts long-term functional recovery in rodent models (*Nielson et al., 2015*). These prior cross-species findings motivated the present multicenter study where we apply a data-driven workflow in patients with ultra-acute SCI surgical records from two different Level 1 trauma centers. By employing machine intelligence tools, we show that deviation from an optimal MAP range during surgery predicts lower likelihood of neurological recovery and suggest new targets for therapeutic intervention.

## Methods

### Retrospective data extraction and cohort selection

Operating room (OR) records from n = 94 patients (98 surgical records, 3 patients with multiple surgeries) from the Zuckerberg San Francisco General Hospital (ZSFG, from 2005 to 2013) and n = 33 patients (33 surgical records) from the Santa Clara Valley Medical Center (SCVMC, from 2013 to

**eLife digest** Spinal cord injury is a devastating condition that involves damage to the nerve fibers connecting the brain with the spinal cord, often leading to permanent changes in strength, sensation and body functions, and in severe cases paralysis. Scientists around the world work hard to find ways to treat or even repair spinal cord injuries but few patients with complete immediate paralysis recover fully.

Immediate paralysis is caused by direct damage to neurons and their extension in the spinal cord. Previous research has shown that blood pressure regulation may be key in saving these damaged neurons, as spinal cord injuries can break the communication between nerves that is involved in controlling blood pressure. This can lead to a vicious cycle of dysregulation of blood pressure and limit the supply of blood and oxygen to the damaged spinal cord tissue, exacerbating the death of spinal neurons. Management of blood pressure is therefore a key target for spinal cord injury care, but so far, the precise thresholds to enable neurons to recover are poorly understood.

To find out more, Torres-Espin, Haefeli et al. used machine learning software to analyze previously recorded blood pressure and heart rate data obtained from 118 patients that underwent spinal cord surgery after acute spinal cord injury.

The analyses revealed that patients who suffered from either low or high blood pressure during surgery had poorer prospects of recovery. Statistical models confirming these findings showed that the optimal blood pressure range to ensure recovery lies between 76 to 104-117 mmHg. Any deviation from this narrow window would dramatically worsen the ability to recover.

These findings suggests that dysregulated blood pressure during surgery affects to odds of recovery in patients with a spinal cord injury. Torres-Espin, Haefeli et al. provide specific information that could improve current clinical practice in trauma centers. In the future, such machine learning tools and models could help develop real-time models that could predict the likelihood of a patient's recovery following spinal cord injury and related neurological conditions.

2015) that underwent spinal surgery were collected retrospectively. For ZSFG, monitoring data was extracted from the values manually recorded by the anesthesiologist at 5 min intervals (Q5). For SCVMC, monitoring data was extracted from the Surgical Information Systems (SIS) (Alpharetta, GA) at 1 min intervals (Q1). Demographics and outcome variables were extracted from an existing retrospective registry. AIS (American Spinal Injury Association [ASIA] Impairment Scale) grade at admission (first complete AIS upon admission to the hospital before surgery) and discharge (latest complete AIS grade after surgery before discharge from hospital) were estimated using the available ISNCSCI exams (International Standards for Neurological Classification of SCI) and the neurosurgery, trauma surgery, emergency department, and physical medicine and rehabilitation physical exam notes. To ensure compatibility between centers on the estimated AIS grades, one independent physician conducted the estimates for all the patients in each center (SM for SCVMC and JT for ZSFG) and one independent ISNCSCI certified physician (WW) extracted the AIS grades for all the patients (across centers). In case of conflict between grades, both physicians established a consensus. From the total 131 surgical records, three records were excluded for not having monitoring recorded for both MAP and HR, three were excluded because surgeries were not related to SCI, and seven surgeries were excluded from three patients that were submitted to more than one surgery. The final cohort for exploratory topological data analysis included 118 patients with complete MAP and heart rate (HR) monitoring. For confirmatory regression analysis, 15 patients were excluded because AIS grade could not be extracted either at admission and/or discharge (*Figure 2—figure supplement 1*). AIS improvement was defined as an increase of at least one AIS grade from admission to discharge. The final list of extracted variables included: MAP and HR continuous monitoring, age, length of surgery in minutes, days from surgery to hospital discharge, estimated AIS grade at admission, estimated AIS grade at discharge and AIS improvement ('yes', 'no'). All data was de-identified before pre-processing and analysis. Protocols for retrospective data extraction were approved by Institutional Research Board (IRB).

**Table 1.** Cohort demographics split by AIS (American Spinal Injury Association [ASIA] Impairment Scale) improvement.

| | AIS improve. N/A (n = 15) | AIS improve. NO (n = 61) | AIS improve. YES (n = 42) | Univariate p-value |
|---|---|---|---|---|
| **Age (years)** | | | | 0.12 |
| Mean (SD) | 46.0 (17.6) | 45.3 (19.1) | 51.4 (19.7) | |
| Median [min, max] | 45.5 [19.0, 87.0] | 47.0 [18.0, 82.0] | 55.0 [18.0, 86.0] | |
| Missing | 1 (6.7%) | 2 (3.3%) | 1 (2.4%) | |
| **AIS admission** | | | | 0.013 |
| A | 1 (6.7%) | 33 (54.1%) | 18 (42.9%) | |
| B | 0 (0%) | 5 (8.2%) | 8 (19.0%) | |
| C | 0 (0%) | 5 (8.2%) | 11 (26.2%) | |
| D | 0 (0%) | 14 (23.0%) | 5 (11.9%) | |
| E | 0 (0%) | 4 (6.6%) | 0 (0%) | |
| Missing | 14 (93.3%) | 0 (0%) | 0 (0%) | |
| **AIS discharge** | | | | <0.0001 |
| A | 0 (0%) | 35 (57.4%) | 0 (0%) | |
| B | 0 (0%) | 5 (8.2%) | 5 (11.9%) | |
| C | 1 (6.7%) | 4 (6.6%) | 15 (35.7%) | |
| D | 0 (0%) | 14 (23.0%) | 17 (40.5%) | |
| E | 1 (6.7%) | 2 (3.3%) | 5 (11.9%) | |
| Missing | 13 (86.7%) | 1 (1.6%) | 0 (0%) | |
| **Surgery duration (min)** | | | | 0.66 |
| Mean (SD) | 433 (167) | 392 (146) | 407 (181) | |
| Median [min, max] | 432 [121, 725] | 389 [120, 728] | 343 [151, 950] | |
| Missing | 1 (6.7%) | 2 (3.3%) | 1 (2.4%) | |
| **Surgery to discharge (days)** | | | | 0.33 |
| Mean (SD) | 9.50 (2.12) | 18.8 (20.6) | 23.4 (23.8) | |
| Median [min, max] | 9.50 [8.00, 11.0] | 11.0 [1.00, 128] | 14.5 [4.00, 120] | |
| Missing | 13 (86.7%) | 4 (6.6%) | 2 (4.8%) | |
| **Dichotomized neurological level of injury at admission** | | | | 0.054 |
| Cervical | 2.00 (13.3%) | 36 (59%) | 33 (78.6%) | |
| Non-cervical | 13.00 (86.7%) | 25 (41%) | 9 (21.4%) | |

## Cohort statistics

The differences in the AIS improvers/non-improvers population characteristics were tested at the univariate level using R (see software below). For continuous numerical variables (age, length of surgery, and days from surgery to discharge), the group mean differences were tested using unpaired Student's t-test (two-sided test). For categorical variables (AIS admission, AIS at discharge, and dichotomized neurological level of injury [NLI]), their levels were compared using Fisher's exact test (two-sided test). p-Values are presented in *Table 1*.

## Topological network extraction and exploration of monitoring data workflow

### Monitoring data pre-processing

Two datasets were generated, one for each center. To ensure compatibility, both datasets were harmonized. Given the difference in the sampling frequency of the monitoring data (Q5 for ZSFG and Q1 SCVMC) between centers and protocols for data collection, SCVMC monitoring data was first pre-processed. Briefly, electronic data was exported from the SIS SQL database, de-identified and imported into MATLAB version 2016b (Mathworks Inc, Natick, MA) for filtering. A custom MATLAB script generated by the SCVMC team implemented filtering criteria established by clinicians and researchers to correct for invalid data (e.g., motion artifacts and injections). Thus, MAP values under 10 and above 200 mmHg as well as point-to-point changes greater than 40 mmHg were filtered, as these instances were found to represent data artifacts. This process was validated by comparing clinical curated data and the extracted data from the script with an accuracy of 99.1 %. After filtering, SCVMC Q1 monitoring data was downsampled to Q5 by taking the average of five consecutive Q1 intervals for compatibility with ZSFG data. Given that the duration of the monitoring for each patient was different and the continuous time-series data is not aligned between patients (without time dependency on monitoring values), the empirical cumulative distribution function (CDF) for each time-series and each patient was computed. To account for the different scales between MAP and HR, a bin width for CDF was set as a 1% of the range of each measure, producing 100 CDF bins for MAP and 100 bins for HR (*Figure 2—figure supplement 2*). Additionally, the average MAP (aMAP) and HR (aHR) across time for each patient was calculated for posterior analysis.

### Similarity between patients

The CDF bins for both MAP and HR were serialized in one vector per patient and the Euclidean distance ($d\left(p,q\right) = \sqrt{\sum_{i=1}^{n}\left(p_i - q_i\right)^2}$, where **p** and **q** are two patients' CDF vectors and $n$ is the number of CDF bins) calculated for each pair of patients' vectors. The Euclidean distance matrix was then processed using dimensionality reduction.

### Dimensionality reduction

Our goal for dimensionality reduction was to increase the signal-to-noise ratio by mapping to a lower number of dimensions ($d$) that contained the major information in the dataset. Dimensionality reduction was achieved by using the Isomap algorithm (*Tenenbaum et al., 2000*). Isomap is a non-linear dimensionality reduction method that uses multidimensional scaling (MDS) with geodesic distances instead of the Euclidean distances as the classic MDS does, and it has been suggested before for health data (*Weng et al., 2005*). Isomap performs three steps: (1) construct an NNG (near neighbor graph), (2) estimate the geodesic distances from the graph (shortest paths), (3) compute MDS embedding with the geodesic distances. The algorithm takes one parameter ($k$ or $e$) to set the threshold for the NNG (we used $k$, the number of near neighbors for NNG). For selecting $k$, we considered the minimal $k$ the smallest one that produced a connected NNG, in our case $k = 3$. Then two criteria were established for both $k$ optimization and $d$ selection, distance preservation (RV, residual variance) and topological persistent homology preservation as described (*Rieck and Leitte, 2015*; *Paul and Chalup, 2017*). We considered Isomap solutions for $k = 3$–7 (*Figure 2—figure supplement 1*). The RV was computed as (*Tenenbaum et al., 2000*): $1 - R^2\left(\hat{D}_m, D_y\right)$ where $R$ is the standard linear correlation coefficient taken over all entries of $\hat{D}_m$ and $D_y$. $\hat{D}_m$ is the input distance matrix that the algorithm is trying to estimate the real dimensions of ($k$-geodesic distance matrix for $k$-Isomap). $D_y$ is the Euclidean distance matrix of the low-dimensional solution. No major differences were observed in RV between the solutions for different $k$, except for the first dimension where RV increases as $k$ increases. Isomap persistence diagrams were obtained using Vietoris-Rips filtration (*Paul and Chalup, 2017*) for $\hat{D}_m$ and $D_y$ for different $d$ solutions (*Figure 2—figure supplement 1*). Then the topological zero-dimensional and one-dimensional Wasserstein (power = 2) distances (WD0 and WD1, respectively) were computed between $\hat{D}_m$ and $D_y$. In persistent homology, dimension 0 measures zero-dimensional *holes* in the data (the connectivity of the datapoints, i.e., the number of connected sets) and topological dimension 1 measures one-dimensional *holes,* namely loops. We sought to select the solution (determine $d$ and $k$)

that minimized the WD0 and WD1, indicative of the optimal solution preserving the major topological information (*Rieck and Leitte, 2015*; *Paul and Chalup, 2017*). Given that $k$ = 6 and $k$ = 7 showed the lowest WD0 and WD1, we considered $k$ = 6 as the final solution (*Figure 2—figure supplement 1*). A $d$ = 4 (four dimensions kept in the $k$ = 6 Isomap) was chosen for being the one at the 'elbow' of the RV, the one that minimized WD0 in $k$ = 6 Isomap and presented a good compromised WD1.

## Network analysis

A network from the $k$ = 6 $d$ = 4 Isomap solution was created for visual representation of the connectivity of patients (similarity) in the low-dimensional space. In this network, nodes represent each patient and edges the connection of two patients that are similar in the Isomap solution. An adjacency (whether two nodes are connected) matrix was obtained by computing a $k$-NNG for the low-dimensional space. The cutoff threshold for adjacency was set to the minimal $k$ that produced a full-connected network. For network clustering, the walktrap algorithm was used (*Pons and Latapy, 2005*) as implemented in the *igraph* R package. Walktrap takes a single parameter, the number of random steps the algorithm uses to determine if nodes are in the same cluster or not. To select the optimal number of steps we computed walktrap solutions of a set of random steps (1–100) and chosen the first solution which maximized modularity ($Q$) as implemented in *igraph* R package (*Clauset et al., 2004*). In network analysis, modularity can be interpreted as the proportion of within cluster compared to the between clusters connectivity (edges). This solution was 47 random steps, producing a dendrogram tree of connectivity which maximal modularity cut the tree in 11 clusters (*Figure 2—figure supplement 1*). Then the network was contracted for visual representation of a cluster network of patients, where nodes represented the clusters and edges connected clusters that had at least one edge in the similarity network. Both the similarity and the cluster networks were used for exploratory network analysis and hypothesis generation by mapping patient features and visual inspection (*Figure 2—figure supplement 1*). We used the assortativity coefficient (Ar) to explore the possibility that the network was capturing the association between patients and the time of MAP out of a range (Figure 4; see time MAP out of range). The Ar was calculated using the *igraph* implementation in R (*Newman, 2003*) and it can be interpreted as the Pearson coefficient (−1–1) between nodes connectivity and value of a variable.

## Regression analysis

### Logistic regression

We first used a logistic regression to model the probability of predicting improvement by aMAP. Visual inspection of the plot (Figure 2) suggested a non-linear relationship between aMAP and the

**Table 2.** Logistic regression likelihood ratio test and leave-one-out cross-validation (LOOCV) error (n = 103 patients).

| Model | AIC | Residual df | Residual deviance | Deviance | p-Value | LOOCV error |
|---|---|---|---|---|---|---|
| Null model ($l = \beta_0$) | 141.26 | 102 | 139.26 | | | 0.246 |
| Linear model ($l = \beta_0 + \beta_1 x$) | 134.8 | 101 | 130.80 | 8.46 (vs. null model) | 0.0036** (vs. null model) | 0.231 |
| Quadratic model ($l = \beta_0 + \beta_1 x + \beta_2 x^2$) | 128.48 | 100 | 122.48 | 8.32 (vs. linear model) | 0.0039** (vs. linear model) | 0.210 |
| Cubic model ($l = \beta_0 + \beta_1 x + \beta_2 x^2 + \beta_3 x^3$) | 126.97 | 99 | 118.97 | 3.50 (vs. quadratic model) | 0.061 (vs. quadratic model) | 0.213 |
| Natural Spline model (df = 2) ($l = \beta_0 + f_1\left(x\right)$) | 128.29 | 100 | 122.29 | 8.50 (vs. linear model) | 0.0035** (vs. linear model) | 0.210 |
| Natural Spline model (df = 3) ($l = \beta_0 + f_1\left(x\right)$) | 127.13 | 99 | 119.13 | 3.34 (vs. quadratic model) | 0.067 (vs. quadratic model) | 0.213 |

** p < 0.01.

**Table 3.** Evaluation of logistic regression (Wald test) and leave-one-out cross-validation (LOOCV) error.

Model: $l = \beta_0 + \beta_1 x_1 + \beta_2 x_1^2$ where $x_1$ : **average MAP (n = 103 patients)**

LOOCV: average observed accuracy = 0.66; average kappa statistic = 0.334

| Predictor | Coef. estimate (logit) | Std. error | z-Value | p-Value |
|---|---|---|---|---|
| Intercept | $\beta_0$= –0.55 | 0.242 | –2.293 | 0.02183* |
| Average MAP ($x_1$) | $\beta_1$= 8.62 | 2.944 | 2.931 | 0.00338** |
| Average MAP ($x_1^2$) | $\beta_2$= –7.601 | 3.039 | –2.501 | 0.0123* |

*p < 0.05; **p < 0.01.

probability of improvement. Consequently, the following logistic models were considered ($l$ being the log-odds or logit of the probability of improving): the null model with no predictors ($l = \beta_0$, the simple model ($l = \beta_0 + \beta_1 x$), the two-grade polynomial model ($l = \beta_0 + \beta_1 x + \beta_2 x^2$), the three-grade polynomial model ($l = \beta_0 + \beta_1 x + \beta_2 x^2 + \beta_3 x^3$) and a natural spline model ($l = \beta_0 + f_1 (x)$ )where $f_1 (x)$ is the natural spline function with 2 or 3 degrees of freedom (df)). The natural cubic spline was chosen to relax the symmetric constraints of polynomial models given that the visual inspection of the data suggested an asymmetric aMAP range (Figure 2, distribution of aMAP of improvers is skewed to the left). The results of fitting these models and the likelihood comparison between them (by likelihood ratio test) are shown in *Table 2*. The best fitting model was the two-grade polynomial (Figure 3) and the natural spline (2df) with significant coefficients, confirming our hypothesis. These results were confirmed by leave-one-out cross-validation (LOOCV) (*Table 2*). To account for the potential confounding effect of AIS grade at admission (given differences between groups, *Table 1*), aHR, length of surgery (minutes), days from surgery to discharge and age, we fitted the quadratic model with those covariates and LOOCV (*Table 3*). Considering the independence of the predictors (small correlation coefficients between variables; *Table 4*), the results of the quadratic term being significant still holds for the covariate model.

## Time out of MAP range

We sought to determine a range of MAP in which time outside that range might predict improvement. To consider the time at which MAP was outside a range, we performed an increasing window of MAP for either a symmetric range or an asymmetric one. For the symmetric range, a 1 mmHg range increment at each site of the center (90 mmHg, the mean MAP for improvers) was created. For the asymmetric range, the lower limit was fixed at 76 mmHg and the upper limit was incremented 1 mmHg at the time. The time of MAP (in min) being outside each range was estimated for each patient.

### LASSO regression

LASSO (least absolute shrinkage and selection operator) regression (*Tibshirani, 1996*) was used for selecting a single range of MAP (see time MAP out of range) predictor of the logistic model: $l = \beta_0 + \sum_{j=1}^{p} \beta_j x_j$, where $x_j$ is the $j$th MAP range. LASSO takes as parameter lambda that sets the amount of shrinkage or regularization (using L1-norm penalty). LOOCV was used to determine the lambda that shrunk the models to one predictor or MAP range. The one-predictor solutions (MAP range of 76–104 for symmetric range model, and 76–117 for the asymmetric range model) were used as the solo predictor of AIS improvers in a logistic regression with LOOCV (see above). It is important to note that given the high multicollinearity in the range data, the Q5 time estimation and the low sample size, the LASSO solution should be taken with caution and as an indicator of the MAP range hypothesis rather than a hard rule for medical decision making.

## Prediction modeling

Logistic regression (see above) was used to build prediction models for three binary outcome metrics: AIS improvement of at least one grade from admission to discharge, whether patient was AIS grade A at discharge, or whether the patient was AIS grade D at discharge. For each one of the classification

**Table 4.** Evaluation of logistic regression with covariates (Wald test) and leave-one-out cross-validation (LOOCV).

Model: $l = \beta_0 + \beta_{11}x_1 + \beta_{12}x_1^2 + \beta_2 x_2 + \beta_3 x_3 + \beta_4 x_4 + \beta_5 x_5 + \beta_6 x_6 + \beta_7 x_7 + \beta_8 x_8 + \beta_9 x_9$, where $x_1$: average MAP; $x_2$: average HR; $x_3$: **length of surgery (min)**; $x_4$: days to AIS discharge (days); $x_5$: age; $x_6$: **AIS admission D ('yes','no')**; $x_7$: **AIS admission C ('yes','no')**; $x_8$: **AIS admission B ('yes','no')**; $x_9$: **AIS admission A ('yes','no')**; (AIS admission E was set as the reference level for AIS admission variable and is part of the intercept) (final $n = 93$)

LOOCV: average observed accuracy = 0.688; average kappa statistic = 0.362

| Predictor | Coef. estimate (logit) | Std. error | z-Value | p-Value |
|---|---|---|---|---|
| Intercept | $\beta_0 = -1.530$ | 121.8 | −0.013 | 0.99 |
| Average MAP ($x_1$) | $\beta_{11} = 7.398$ | 3.112 | 2.377 | 0.017* |
| Average MAP ($x_1^2$) | $\beta_{12} = -8.053$ | 3.530 | −2.281 | 0.022* |
| Average HR ($x_2$) | $\beta_2 = -2.087$ | 0.0245 | −0.851 | 0.394 |
| Length of surgery ($x_3$) | $\beta_3 = 0.0011$ | 0.0015 | 0.728 | 0.466 |
| Days to AIS discharge ($x_4$) | $\beta_4 = 0.0037$ | 0.0109 | 0.344 | 0.730 |
| Age ($x_5$) | $\beta_5 = 0.0082$ | 0.013 | 0.634 | 0.526 |
| AIS admission D ($x_6$) | $\beta_6 = 1.454$ | 1.218 | 0.012 | 0.990 |
| AIS admission C ($x_7$) | $\beta_7 = 1.645$ | 1.218 | 0.014 | 0.989 |
| AIS admission B ($x_8$) | $\beta_8 = 1.585$ | 1.218 | 0.013 | 0.989 |
| AIS admission A ($x_9$) | $\beta_9 = 1.527$ | 1.218 | 0.013 | 0.990 |

Correlation matrix (Spearman)

| | Average MAP | Average HR | Length of surgery | Days to AIS discharge | Age | AIS admission |
|---|---|---|---|---|---|---|
| Average MAP | 1 | | | | | |
| Average HR | −0.126 | 1 | | | | |
| Length of surgery | −0.152 | 0.101 | 1 | | | |
| Days to AIS discharge | 0.088 | −0.059 | 0.165 | 1 | | |
| Age | 0.006 | −0.245 | 0.011 | 0.022 | 1 | |
| AIS admission | 0.024 | 0.003 | −0.01 | 0.258 | −0.13 | 1 |

*p < 0.05.

tasks, the following predictors were considered: quadratic aMAP (both linear and quadratic terms), aHR, length of surgery (min), days from surgery to discharge, age, AIS grade at admission, and dichotomized NLI. We performed model selection (a.k.a. feature selection) through an exhaustive search of all potential combinations of at least one of the predictors using the *glmulti* R package (*Calcagno, 2020*). The most parsimonious models were selected to be the one minimizing the small-sample corrected Akaike information criteria (AIC) for each task. We then investigated the performance of each one of the most parsimonious models using LOOCV and adjusting the classification threshold to balance prediction sensitivity and specificity. Briefly, each model was trained $n$ (patient) times with an

$n-1$ training sample and tested the performance in the remaining sample. A vector of $n$ probabilities of predictions was then used to measure the LOOCV model performance. Model fitting and prediction performance were conducted using the *caret* R package (*Kuhn et al., 2019*). Receiving operating curves (ROC) and area under the curve (AUC) for the LOOCV prediction were obtained using the ROCR R package (*Sing et al., 2005*).

## Software

All data wrangling, processing, visualization, and analysis was performed using the R programming language (R version 3.5.1) (*R core team, 2019*) and RStudio (RStudio version 1.2.1335) (*Team, 2018*) in Windows 10 operating system, with the exception of the Q1 OR measures form SCVMC that were preprocessed in MATLAB before downsampling to Q5 in R. The most relevant R functions and packages (beyond the installed with R) used and the references for each function/package and methods are reported in the following table. For more details, see the source code available (*Supplementary file 1* and *Source code 1*).

## R packages used

| Package | Version | Usage | Reference |
|---|---|---|---|
| *igraph* | 1.2.4.1 | Building, manage and analyze networks | *Csárdi and Nepusz, 2006* |
| *dplyr* | 0.8.3 | Data cleaning and wrangling | *Wickham et al., 2018* |
| *ggplot2* | 3.2.1 | Data visualization and plotting | *Wickham, 2016* |
| *vegan* | 2.5–5 | For Isomap implementation | *Jari et al., 2019* |
| *RColorBrewer* | 1.1–2 | To control and create colormaps | *Neuwirth, 2014* |
| *TDAstats* | 0.4.0 | Utilities for topology data analysis for persistent homology | *Wadhwa et al., 2018* |
| *cccd* | 1.5 | For generating NNGs | *Marchette, 2015* |
| *table1* | 1.1 | Generates table of demographics | *Rich, 2018* |
| *glmnet* | 2.0–18 | For fitting LASSO | *Friedman et al., 2010* |
| *glmnetUtils* | 1.1.2 | For fitting LASSO | *Ooi, 2019* |
| *caret* | 6.0–84 | To fit logistic regression with leave-one-out cross-validation | *Kuhn et al., 2019* |
| *splines* | 3.5.1 | To fit the spline models | *R core team, 2019* |
| *VisNetwork* | 2.0.7 | Visualization suit for network graphs using the vis. js JavaScript library | *Almende, 2019* |
| *stats* | 3.5.1 | Fit generalized linear models | *R core team, 2019* |
| *glmulti* | 1.0.8 | For model search | *Calcagno, 2020* |
| *ROCR* | 1.0–11 | For ROC visualization and performance | *Sing et al., 2005* |
| *reshape2* | 1.4.3 | From wide to long view dataframe formatting | *Wickham, 2007* |

## Data and code availability

The final de-identified datasets for analysis are deposited and accessible at the Open Data Commons for SCI (odc-sci.org, RRID:SCR_016673) under DOIs 10.34945 /F5R59J and 10.34945 /F5MG68. The R code to run all the analysis present in this publication, including visualizations, is available as supplementary material. The code would reproduce the entire analysis and plots when run using the same versions of R, RStudio, and packages specified in this publication. Otherwise results might change.

## Results

### Exploratory network analysis suggests an upper and lower limit of intra-operative MAP for recovery

Intra-operative monitoring records (MAP, HR) and neurological outcome data were extracted and curated from two Level 1 trauma centers. A final cohort of 118 patients was included (*Figure 1a* and *Table 1*). The cohort represents a varied dataset of intra-operative MAP and HR patterns and respective aMAP across time in surgery and aHR across time in surgery values (*Figure 1b–c*). Using a machine intelligence analytical pipeline (*Figure 2a*), we extracted a similarity network of patients (*Pai and Bader, 2018*; *Parimbelli et al., 2018*) from a low-dimensional space embedded using a non-linear algorithm, Isomap (*Tenenbaum et al., 2000*), on a distance matrix derived from the MAP and HR records and then performed topological network extraction using persistent homology metrics (*Rieck and Leitte, 2015*; *Figure 2a and Figure 2—figure supplement 1*). The results of this dimensionality reduction suggested that four dimensions are enough to capture most of the variance and the topological structures of the original data (*Figure 2—figure supplement 1c-e*). Clustering the network of patients through a random-walk algorithm, Walktrap (*Pons and Latapy, 2005*), revealed 11 different clusters where patients were regarded to share intra-operative hemodynamic phenotypes (*Figure 2 and Figure 2—figure supplement 1f-h*). Importantly, this workflow was unsupervised: only the OR hemodynamic time-series was used to derive patient clustering, and therefore any association captured by the network must be dependent on hemodynamic patterns. Exploratory network analysis showed a gradient distribution of patients by their aMAP (*Figure 2b–d*) and aHR (*Figure 2e–g*), confirming that the network captured a valid representation of the raw high-dimensional dataset. We then investigated the association of the clusters to patient recovery as defined by whether the patient improved at least one AIS grade A–D (*Roberts et al., 2017*) between time before surgery and time of discharge from the hospital. Mapping the proportion of patients with AIS improvement onto the similarity network (*Figure 2h–j*) revealed that patients with recovery localized to clusters associated with a middle range of MAP (*Figure 2k*). Those clusters also showed a higher proportion of less severe AIS grades at discharge (AIS C, D, and E) than other clusters (*Figure 2—figure supplement 2*). In contrast, clusters of patients showing an extreme variation of MAP were highly enriched with patients with no AIS recovery and patients with more severe AIS grades at discharge (AIS A and B, *Figure 2—figure supplement 2*). This analysis suggested that there is a limited range of MAP during surgery associated with neurological recovery.

### MAP has a non-linear association with probability of recovery

The exploratory network analysis revealed that clusters with higher proportion of patients that increased AIS of at least one grade were associated with having a middle range aMAP (*Figure 2 and Figure 2—figure supplements 1 and 2*) and that clusters of patients with aMAP on the extremes contained fewer improvers. We hypothesized that there might be a non-linear relationship between intra-operative MAP and the probability of AIS grade improvement. To confirm this hypothesis, logistic regression models with LOOCV were used (*Figure 3*, *Table 2*). We fitted a null model (no predictors) as well as linear, polynomial, and cubic models of aMAP (*Figure 3*, *Table 2*) to test the non-linearity of the hypothesis. The linear model showed a significant improvement over the null model with a positive association, suggesting that the higher the aMAP, the higher the probability of AIS grade improvement. However, polynomial logistic regression demonstrated a significant quadratic fit (*Table 2*) with lower LOOCV error than the linear model, indicating that a quadratic form of aMAP better predicts the probability of improvement. Notably, the cubic model did not show significant improvement over the quadratic one. Exploratory network analysis suggested an asymmetrical function of AIS improvement with respect to aMAP (*Figure 2k*); we therefore also tested spline models to relax the symmetry of polynomial models. A spline model of degree 2 resulted in a significant fit over the linear model (*Table 2*) while a spline model of degree 3 resulted in a similar fit as compared to the cubic model. There was no evidence from which to choose between the spline model of degree 2 and the quadratic model. Accordingly, we did not pursue the asymmetric model further, although we explore an asymmetric MAP range below.

### Factors influencing MAP association with recovery

We sought to explore additional patient characteristics that might explain or affect MAP association with recovery. To test whether other factors could be responsible for the observed non-linear

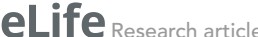

**Figure 1.** High-frequency monitoring operating room (OR) data. Flowchart of retrospective study and cohort selection criteria (**a**). A final cohort of 118 patients were identified and values of mean arterial pressure (MAP) (**b**) and heart rate (HR), (**c**) by time (bins of 5 min; **Q5**) retrospectively extracted from patients' records. Colormaps represent the MAP (mmHg; green marks normotensive MAP, while blue and red marks hypotension and hypertension, respectively) and HR (beats per min, bpm; dark yellow lowest to purple highest) at each Q5 time, depicting the temporal fluctuation of each measure for each patient (row). The average MAP (aMAP, right plot in **b**) and average heart rate (aHR, right plot in **c**) were computed.



**Figure 2.** Topological network analysis of intra-operative monitoring. Intra-operative mean arterial pressure (MAP) and heart rate (HR) sampled every 5 min (**Q5**) were curated, processed, and formatted in a unique data matrix (**a**) (**Figure 2—figure supplement 1**). The similarity matrix between patients was computed and a four-dimensional subspace extracted using Isomap (**Figure 2—figure supplement 1**). A network was constructed where nodes represent patients and edges the connection of pairs of patients under a specified threshold of similarity (see Methods). The network was clustered and

*Figure 2 continued on next page*

*Figure 2 continued*

collapsed (***Figure 2—figure supplement 1***) by using the walktrap algorithm conveying in 11 clusters. These networks captured both the average MAP (aMAP) (**b–d**) and the average HR (aHR) (**e–g**) in a gradient fashion. Similarly, at least one AIS grade gain at discharge ('yes', 'no') was mapped over the network (**h–j**, gray: 15 AIS grades could not be extracted). Clusters of higher proportion of patients with recovery had an aMAP in a middle range, while clusters with higher proportion of patients without recovery presented extreme aMAPs (**k**). The mean cluster aHR showed a less apparent relationship with the proportion of AIS improvers (**l**).

The online version of this article includes the following figure supplement(s) for figure 2:

**Figure supplement 1.** Data preprocessing, dimensionality reduction, and clustering optimization.

**Figure supplement 2.** Exploratory network analysis of AIS (American Spinal Injury Association [ASIA] Impairment Scale) at discharge.

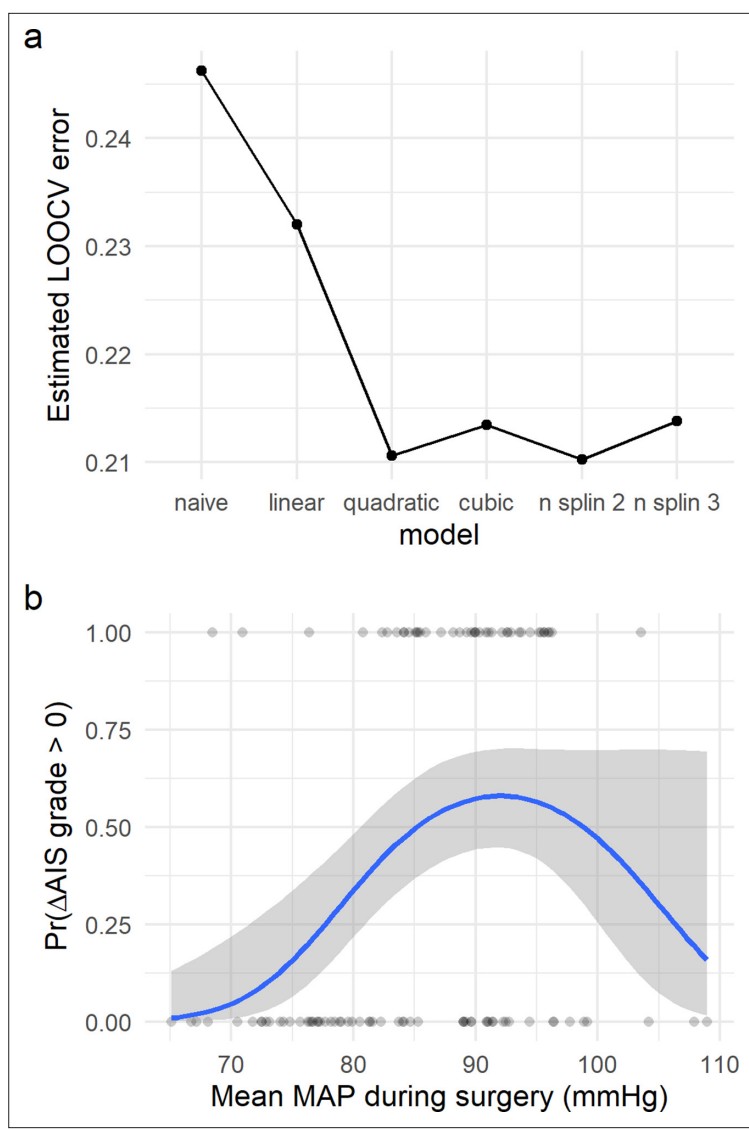

**Figure 3.** Non-linear relationship of average mean arterial pressure (aMAP) with the probability of improving at least one AIS grade. Logistic regression models were fitted to study the potential non-linearity of the aMAP predictor as suggested by the exploratory analysis. Six different models were studied: naïve, linear, quadratic, cubic, spline of degree 2, and spline of degree 3. The estimated leave-one-out cross-validation (LOOCV) error for each model showed that both the quadratic and the spline of degree 2 have the minimal cross-validation error (**a**). This suggests that the linear model did not capture all the potential explainable variance of the response variable by aMAP, while the cubic and spline of degree 3 were probably overfitting the model (Table 2). (**b**) shows the logit function (blue line) and standard error (gray ribbon) of the quadratic model over the fitted values (points).

**Table 5.** Evaluation of logistic regression covariates only (Wald test) and leave-one-out cross-validation (LOOCV).

Model: $l = \beta_0 + \beta_2 x_2 + \beta_3 x_3 + \beta_4 x_4 + \beta_5 x_5 + \beta_6 x_6 + \beta_7 x_7 + \beta_8 x_8 + \beta_9 x_9$, where $x_2$: average HR; $x_3$: length of surgery (min); $x_4$: days to AIS discharge (days); $x_5$: age; $x_6$: AIS admission D ('yes','no'); $x_7$: AIS admission C ('yes','no'); $x_8$: AIS admission B ('yes','no'); $x_9$: AIS admission A ('yes','no'); (AIS admission E was set as the reference level for AIS admission variable and is part of the intercept) (final $n = 93$)

LOOCV: average observed accuracy = 0.612; average kappa statistic = 0.17

| Predictor | Coef. estimate (logit) | Std. error | z-Value | p-Value |
|---|---|---|---|---|
| Intercept | $\beta_0 = -1.585$ | 138.2 | −0.011 | 0.991 |
| Average HR ($x_2$) | $\beta_2 = -0.0209$ | 0.029 | −0.911 | 0.362 |
| Length of surgery ($x_3$) | $\beta_3 = 0.0016$ | 0.00141 | 1.151 | 0.250 |
| Days to AIS discharge ($x_4$) | $\beta_4 = 0.0105$ | 0.0106 | 0.993 | 0.320 |
| Age ($x_5$) | $\beta_5 = 0.0052$ | 0.012 | 0.424 | 0.672 |
| AIS admission D ($x_6$) | $\beta_6 = 1.511$ | 1.382 | 0.011 | 0.991 |
| AIS admission C ($x_7$) | $\beta_7 = 1.715$ | 1.382 | 0.012 | 0.991 |
| AIS admission B ($x_8$) | $\beta_8 = 1.643$ | 1.382 | 0.012 | 0.990 |
| AIS admission A ($x_9$) | $\beta_9 = 1.574$ | 1.382 | 0.011 | 0.991 |

association, we first compared the quadratic model with aMAP as a predictor alone, a model that also includes several covariates (aHR, length of surgery, days from surgery to discharge, age, and AIS grade at admission), and a model with only the covariates. The significance of the quadratic fit holds even after accounting for the covariates (**Table 3, 4**), and none of the terms in the covariates-only model had a significant coefficient (**Table 5**). These results indicate that even in the presence of the other factors, aMAP is still non-linearly associated with AIS grade improvement at discharge.

Patients with more severe injuries are more likely to suffer hemodynamic dysregulations (**Lehmann et al., 1987**). Hence, we studied whether the relationship of MAP and AIS improvement was maintained in the subcohort of patients with an AIS grade of A at admission. We first filtered the data for

**Table 6.** Evaluation of logistic regression in American Spinal Injury Association (ASIA) Impairment Scale (AIS) A at admission cohort (Wald test) and leave-one-out cross-validation (LOOCV).

Model: $l = \beta_0 + \beta_{11} x_1 + \beta_{12} x_1^2 + \beta_2 x_2 + \beta_3 x_3 + \beta_4 x_4 + \beta_5 x_5$, where $x_1$: average MAP; $x_2$: average HR; $x_3$: length of surgery (min); $x_4$: days to AIS discharge (days); $x_5$: age (final $n = 51$)

LOOCV: average observed accuracy = 0.63; average kappa statistic = 0.197

| Predictor | Coef. estimate (logit) | Std. error | z-Value | p-Value |
|---|---|---|---|---|
| Intercept | $\beta_0 = -0.931$ | 3.433 | −0.271 | 0.786 |
| Average MAP ($x_1$) | $\beta_{11} = 10.79$ | 5.014 | 2.153 | 0.031* |
| Average MAP ($x_1^2$) | $\beta_{12} = -6.73$ | 4.591 | −1.468 | 0.142 |
| Average HR ($x_2$) | $\beta_2 = -0.016$ | 0.035 | −0.468 | 0.639 |
| Length of surgery ($x_3$) | $\beta_3 = 0.0039$ | 0.0026 | 1.504 | 0.132 |
| Days to AIS discharge ($x_4$) | $\beta_4 = 0.0067$ | 0.014 | 0.477 | 0.633 |
| Age ($x_5$) | $\beta_5 = -0.012$ | 0.020 | −0.599 | 0.549 |

*p < 0.05.

**Table 7.** Neurological level of injury cases.

| | Cervical (n = 71) | | | | | | | Non-cervical (n = 32) | | | | | | | | | | | | |
|---|---|---|---|---|---|---|---|---|---|---|---|---|---|---|---|---|---|---|---|---|
| NLI | C2 | C3 | C4 | C5 | C6 | C7 | C8 | T2 | T3 | T4 | T5 | T6 | T7 | T8 | T9 | T10 | T11 | T12 | S1 | S5 |
| Cases | 3 | 3 | 24 | 28 | 4 | 8 | 1 | 1 | 3 | 3 | 1 | 2 | 1 | 3 | 1 | 3 | 2 | 4 | 6 | 2 |

the subcohort and then fitted a full model as above but without the AIS grade at admission covariate. The resulting model showed the linear aMAP coefficient to be significant and the quadratic term close to significant (p = 0.14) with the second biggest coefficient (*Table 6*). A likelihood ratio test between a linear model with covariates and a quadratic model with covariates resulted in p-values = 0.07. On the other hand, in the full model with covariates fitted to the entire cohort, none of the AIS grades at admission had significant coefficients, which suggested that the non-linear relationship of MAP with neurological recovery was sustained across injury severity in that model. This apparent divergence in results might be explained by the reduction in power for the AIS A cohort model.

Next, given that the level of the cord injury can be related to different degrees of hemodynamic dysregulation (*Lehmann et al., 1987*), we studied the effect of the NLI at admission on the association of MAP and patient recovery. Our cohort was very heterogeneous on the NLI, with most patients having cervical injuries and the rest distributed along the mid and lower segments of the cord (*Table 7*). Thus, we divided the population into two categories: cervical and non-cervical patients. Running the same full model with just the cervical patients resulted in similar results as compared to the full model on the entire cohort, maintaining the quadratic aMAP significance (*Table 8*). In the non-cervical cohort, we did not find a significant association of the quadratic aMAP to recovery (*Table 9*). We then performed additional analyses to determine whether this difference in aMAP relationship to recovery between cervical and non-cervical patients was due to differences in the likelihood of recovery between the two NLI populations. A univariate analysis suggested that the proportion of improvers and not improvers in the cervical and non-cervical population were marginally different

**Table 8.** Evaluation of logistic regression in Cervical cohort (Wald test) and leave-one-out cross-validation (LOOCV).

Model: $l = \beta_0 + \beta_{11}x_1 + \beta_{12}x_1^2 + \beta_2 x_2 + \beta_3 x_3 + \beta_4 x_4 + \beta_5 x_5 + \beta_6 x_6 + \beta_7 x_7 + \beta_8 x_8$, where $x_1$: **average MAP;** $x_2$: **average HR;** $x_3$: **length of surgery (min);** $x_4$: **days to AIS discharge (days);** $x_5$: **age;** $x_6$: **AIS admission D ('yes','no');** $x_7$: **AIS admission C ('yes','no');** $x_8$: **AIS admission B ('yes','no'); (AIS admission A was set as the reference level for AIS admission variable and is part of the intercept, no AIS admission E was present in this cohort) (final n = 93)**

LOOCV: average observed accuracy = 0.688; average kappa statistic = 0.362

| Predictor | Coef. estimate (logit) | Std. error | z-Value | p-Value |
|---|---|---|---|---|
| Intercept | $\beta_0$= 2.747 | 3.018 | 0.91 | 0.363 |
| Average MAP ($x_1$) | $\beta_{11}$= 7.594 | 3.056 | 2.485 | 0.013* |
| Average MAP ($x_1^2$) | $\beta_{12}$= –7.528 | 3.358 | –2.242 | 0.025* |
| Average HR ($x_2$) | $\beta_2$= –0.055 | 0.034 | –1.608 | 0.108 |
| Length of surgery ($x_3$) | $\beta_3$= 0.0014 | 0.0019 | 0.720 | 0.472 |
| Days to AIS discharge ($x_4$) | $\beta_4$= 0.0022 | 0.012 | 0.182 | 0.855 |
| Age ($x_5$) | $\beta_5$= 0.0079 | 0.016 | 0.482 | 0.630 |
| AIS admission D ($x_6$) | $\beta_6$= –0.747 | 0.87 | –0.840 | 0.730 |
| AIS admission C ($x_7$) | $\beta_7$= 0.745 | 0.80 | 0.925 | 0.355 |
| AIS admission B ($x_8$) | $\beta_8$= 0.301 | 0.88 | 0.346 | 0.401 |

*p < 0.05.

**Table 9.** Evaluation of logistic regression in non-cervical cohort only (Wald test) and leave-one-out cross-validation (LOOCV).

Model: $l = \beta_0 + \beta_{11}x_1 + \beta_{12}x_1^2 + \beta_2x_2 + \beta_3x_3 + \beta_4x_4 + \beta_5x_5 + \beta_6x_6 + \beta_7x_7 + \beta_8x_8 + \beta_9x_9$, where $x_1$: average MAP; $x_2$: average HR; $x_3$: length of surgery (min); $x_4$: days to AIS discharge (days); $x_5$: age; $x_6$: AIS admission D ('yes','no'); $x_7$: AIS admission C ('yes','no'); $x_8$: AIS admission B ('yes','no'); $x_9$: AIS admission A ('yes','no'); (AIS admission E was set as the reference level for AIS admission variable and is part of the intercept) (final $n = 93$)

LOOCV: average observed accuracy = 0.688; average kappa statistic = 0.362

| Predictor | Coef. estimate (logit) | Std. error | z-Value | p-Value |
|---|---|---|---|---|
| Intercept | $\beta_0 = -1.883$ | 352.4 | −0.005 | 0.996 |
| Average MAP ($x_1$) | $\beta_{11} = -0.206$ | 4.713 | −0.044 | 0.965 |
| Average MAP ($x_1^2$) | $\beta_{12} = -8.064$ | 7.643 | −1.055 | 0.291 |
| Average HR ($x_2$) | $\beta_2 = -0.0002$ | 0.0649 | 0.004 | 0.997 |
| Length of surgery ($x_3$) | $\beta_3 = 0.0018$ | 0.0054 | 0.336 | 0.737 |
| Days to AIS discharge ($x_4$) | $\beta_4 = 0.076$ | 0.0613 | 1.240 | 0.215 |
| Age ($x_5$) | $\beta_5 = -0.0047$ | 0.051 | −0.921 | 0.357 |
| AIS admission D ($x_6$) | $\beta_6 = 1.727$ | 3.524 | 0.005 | 0.996 |
| AIS admission C ($x_7$) | $\beta_7 = 3.557$ | 5.782 | 0.005 | 0.996 |
| AIS admission B ($x_8$) | $\beta_8 = 1.738$ | 3.524 | 0.005 | 0.995 |
| AIS admission A ($x_9$) | $\beta_9 = 1.686$ | 3.524 | 0.005 | 0.996 |

(*Table 1*). Moreover, a logistic regression predicting AIS grade improvement by NLI categorization indicated that non-cervical patients were significantly less likely to recover ($\beta = -0.93$, p = 0.041). While these results suggest that a quadratic aMAP is important for predicting AIS grade recovery in cervical patients, the lack of significant results in the non-cervical patients must be interpreted with caution due to the reduced number of cases, the heterogeneous distribution, and the low number of improvers in the group.

Finally, we sought to determine whether the probability of recovery associated to MAP could be influenced by the time the patient is in the hospital. For that, we break down the potential causal pathway between MAP dysregulation, AIS improvement, and days from surgery to discharge. We first fitted a logistic regression model with AIS improvement as response and days to discharge as the only predictor. This resulted in a non-significant p-value of p = 0.32, suggesting that days to discharge does not associate with probability of improvement. Second, we fitted a linear model with days to discharge as a response and quadratic aMAP (both linear and quadratic terms) as predictors. This resulted as a significant coefficient of the quadratic term (p = 0.047), although the model was not significant (p = 0.13 for the F statistic) and the adjusted $R^2$ was small (0.0217). We also investigated whether days to discharge interacts with MAP and quadratic MAP to predict AIS improvement, with no significant results on the interaction (interaction days to discharge with aMAP: linear term p = 0.61; quadratic term p = 0.91). These suggest that these two factors do not moderate each other. Finally, eliminating days to discharge from the full covariate model predicting AIS improvement does not have a major effect on the model fit. A likelihood ratio test between both models shows a non-significant change in variance explained (p = 0.729) with a deviance difference of ~0.1 %. All together indicates that the non-linear relationship between aMAP and AIS improvement is independent of the days from surgery to discharge.

## Intra-operative MAP range from 76-[104-117] mmHg predicts recovery

Since aMAP can obscure episodes of high deviation from average (*Hawryluk et al., 2015*) and has a non-linear relationship with recovery, we hypothesized that there might be a range of intra-operative

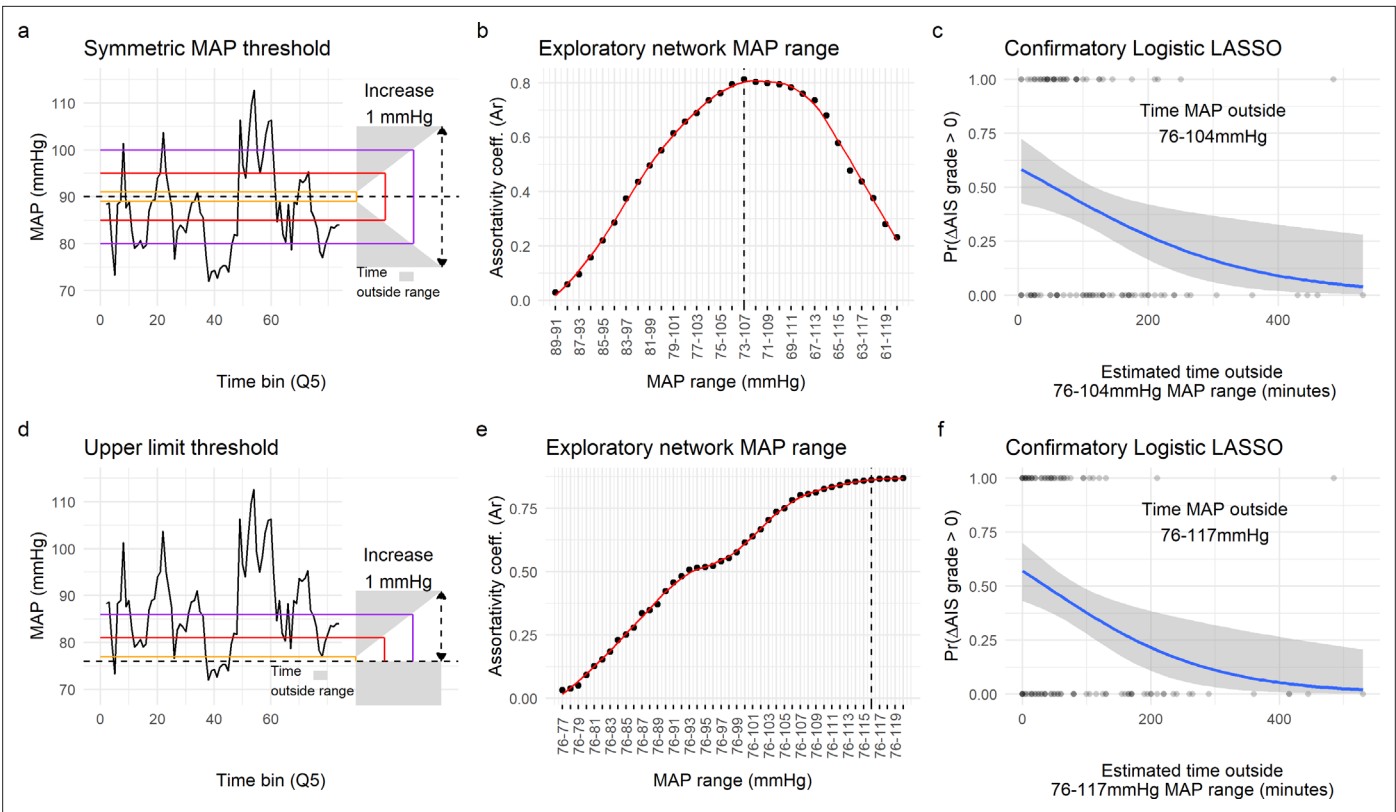

**Figure 4.** Range of mean arterial pressure (MAP). To find the optimal MAP range, a moving MAP range was computed and the time of MAP outside range calculated (**a** and **d**, example of the same patient for symmetric and asymmetric map range, respectively). Calculating the assortativity coefficient (Ar) of the network for each range revealed that the distribution of patients in the network was most highly associated with the range 73–107 mmHg for the symmetric range (**b**, *Figure 4—figure supplement 1*), and 76–116 mmHg for the asymmetric range study of upper limit threshold (**e**, *Figure 4—figure supplement 1*). A logistic LASSO (least absolute shrinkage and selection operator) regression (*Figure 4—figure supplement 2a*, *Figure 4—figure supplement 3*) was used as a confirmatory analysis and to obtain the MAP range that most highly predicts AIS grade recovery. For the symmetric range, the time of MAP outside the 76–104 mmHg (**c**, Table 10) was found to be the 'last-standing' predictor during LASSO regularization (*Figure 4—figure supplement 2a*), suggesting that greater duration outside this range is associated with lower probability of neurological recovery. In the case of the asymmetric range (*Figure 4—figure supplement 3*), the last non-zero coefficient was for the range 76–117 mmHg (**f**, Table 11).

The online version of this article includes the following figure supplement(s) for figure 4:

**Figure supplement 1.** Exploratory network analysis of mean arterial pressure (MAP) out of range.

**Figure supplement 2.** Logistic least absolute shrinkage and selection operator (LASSO) regression with leave-one-out cross-validation (LOOCV) of symmetric range.

**Figure supplement 3.** Logistic least absolute shrinkage and selection operator (LASSO) regression with leave-one-out cross-validation (LOOCV) of asymmetric range.

MAP that better predicts AIS grade improvers. To test this hypothesis, we analyzed the amount of time MAP was out of a specific range (*Figure 4*). Since our modeling suggested both a symmetric and asymmetric range, we performed two different analyses. First, starting at a MAP of 90 mmHg, we implemented an algorithm to iteratively expand the MAP range symmetrically 1 mmHg higher and lower and calculate the time MAP was outside the range (*Figure 4a*). Exploratory analysis of the similarity network indicated a high association between the time out of a MAP range of 73–107 mmHg with the topological distribution of patients (*Figure 4b and Figure 4—figure supplement 1*). To validate this range and the associated lower and upper MAP thresholds, we used a logistic model with LASSO regularization with the predictors being the time outside of each MAP range as in *Figure 4b*. This allowed us to systematically reduce the number of relevant predictors until only one remained (non-zero coefficient). Interestingly, the logistic LASSO regression with LOOCV revealed that a MAP range from 76 to 104 mmHg was optimal in our dataset since it produced the most reproducible prediction of recovery (average LOOCV prediction accuracy of 61.16 %; *Figure 4c and Figure 2—figure*

**Table 10.** Least absolute shrinkage and selection operator (LASSO) solution and logistic regression of most predictive symmetric range with leave-one-out cross-validation (LOOCV).

Model: $l = \beta_0 + \beta_1 x_1$, where $x_1$: time of MAP outside range 76–104 mmHg (*n* = 103 patients)

LOOCV: average observed accuracy = 0.61; average kappa statistic = 0.158

| Predictor | Coef. estimate (logit) | Std. error | z-Value | p-Value |
|---|---|---|---|---|
| Intercept | $\beta_0$ = 0.368 | 0.333 | 1.103 | 0.269 |
| Time MAP out 76–104 ($x_1$) | $\beta_1$ = –0.006 | 0.0026 | –2.566 | 0.0103* |

*p < 0.05.

supplement 2, *Table 9*). Next, we studied the possibility of an asymmetric range by fixing the lower limit to 76 mmHg and increasing the upper limit by 1 mmHg at the time (*Figure 4d*). The association of the patient distribution in the network plateau at a range of 76–116 mmHg (*Figure 4e and Figure 4—figure supplement 1*) and the logistic LASSO found the range 76–117 mmHg be the most predictive of recovery (average cross-validation prediction accuracy of 57.28 %; *Figure 4f and Figure 4—figure supplement 2a*, *Table 10*). While both the exploratory analysis and the logistic LASSO produced similar ranges, the later analysis is performed through statistical modeling rather than descriptive associations, and therefore we further discuss the results of the LASSO.

Altogether, the findings indicate that the time of MAP outside a measurable normotensive range during surgery is associated with lower odds of recovering at least one AIS grade. Our analysis suggests the optimal range for recovery is between 76–104 and 76–117 mmHg. Notice that while range 76–104 mmHg has higher predictive utility than 76–117 mmHg (mean LOOCV accuracy of 61.16 % vs. 57.28%), the difference in variance of the probability of AIS improvement explained by these two predictors is minimal (<4% difference in RV). Therefore, from a modeling perspective, we broadly conclude that the upper limit of the MAP range is probably anywhere between 104 and 117 mmHg.

## Building a predictive model of outcome

Finally, we wanted to study the prediction utility of a model including the analyzed features together with other patient characteristics. We focused on three classification tasks: a model predicting AIS improvement of at least one grade at discharge, a model predicting AIS A at discharge, and a model predicting AIS D at discharge. We chose to predict AIS A and D instead of a multiclass prediction of the AIS at discharge in concordance to our previous studies (*Kyritsis et al., 2021*) and because of the low representation of other grades in our dataset (*Table 1*). For each of the three classification tasks, we performed an exhaustive search of all possible additive models with at least one of the predictors of interest: quadratic aMAP, aHR, length of surgery, days from surgery to discharge, age, AIS grade at admission, dichotomized NLI (cervical, non-cervical), time of MAP out of range 76–104, and time of MAP out of range 76–117. We selected the parsimonious model as the model that minimized the small-sample corrected AIC (*Table 12*). Next, for the selected best model for each task, we

**Table 11.** Least absolute shrinkage and selection operator (LASSO) solution and logistic regression of most predictive asymmetric range with leave-one-out cross-validation (LOOCV).

Model: $l = \beta_0 + \beta_1 x_1$, where $x_1$: time of MAP outside range 76–117 mmHg (*n* = 103 patients)

LOOCV: average observed accuracy = 0.5728; average kappa statistic = 0.102

| Predictor | Coef. estimate (logit) | Std. error | z-Value | p-Value |
|---|---|---|---|---|
| Intercept | $\beta_0$ = 0.2881 | 0.287 | 1.002 | 0.316 |
| Time MAP out 76–117 ($x_1$) | $\beta_1$ = –0.00788 | 0.0027 | –2.828 | 0.00468** |

**p < 0.01.

**Table 12.** Best prediction models of outcome.

**Model predicting AIS improvement:**
$l = \beta_0 + \beta_{11}x_1 + \beta_{12}x_1^2 + \beta_2 x_2 + \beta_3 x_3 + \beta_4 x_4 + \beta_5 x_5$
**Model predicting AIS A:**
$l = \beta_0 + \beta_{11}x_1 + \beta_{12}x_1^2 + \beta_2 x_2 + \beta_3 x_3 + \beta_4 x_4 + \beta_5 x_5 + \beta_6 x_6 + \beta_7 x_7$
**Model predicting AIS D:**
$l = \beta_0 + \beta_2 x_2 + \beta_3 x_3 + \beta_4 x_4 + \beta_5 x_5 + \beta_8 x_8 + \beta_9 x_9$
where $x_1$: average MAP; $x_2$: AIS admission A ('yes', 'no'); $x_3$: AIS admission B ('yes', 'no'); $x_4$: AIS admission C ('yes', 'no'); $x_5$: AIS admission D ('yes', 'no'); $x_6$: NLI non-cervical; $x_7$: Time MAP out 76–117; $x_8$: Length of surgery; $x_9$: Age; (AIS admission E and NLI cervical were set as the reference levels for the corresponding variable and are part of the intercept). All metrics are on LOOCV prediction ($n$ = 93)

| | Model AIS improv. | Model AIS A | Model AIS D |
| --- | --- | --- | --- |
| Predictor | Coef. estimate (logit) | Coef. estimate (logit) | Coef. estimate (logit) |
| Intercept | $\beta_0 = -16.24$ | $\beta_0 = 20.466$ | $\beta_0 = 1.558$ |
| Average MAP ($x_1$) | $\beta_{11} = 7.374$ | $\beta_{11} = 27.031$ | |
| Average MAP (*Cohn et al., 2010*) ($x_1$) | $\beta_{12} = -8.215$ | $\beta_{12} = -17.138$ | |
| AIS admission A ($x_2$) | $\beta_2 = 15.54$ | $\beta_2 = -22.814$ | $\beta_2 = 2.324$ |
| AIS admission B ($x_3$) | $\beta_3 = 16.1818$ | $\beta_3 = -20.38$ | $\beta_3 = 0.41$ |
| AIS admission C ($x_4$) | $\beta_4 = 16.752$ | $\beta_4 = -19.01$ | $\beta_4 = -2.591$ |
| AIS admission D ($x_5$) | $\beta_5 = 14.828$ | $\beta_5 = 0.217$ | $\beta_5 = -2.624$ |
| NLI non-Cervical ($x_6$) | | $\beta_6 = -1.228$ | |
| Time MAP out 76–117 ($x_7$) | | $\beta_7 = 0.017$ | |
| Length of Surgery ($x_8$) | | | $\beta_8 = -0.0044$ |
| Age ($x_9$) | | | $\beta_9 = 0.03$ |
| Model performance metric | Metric value | Metric value | Metric value |
| Accuracy (95% CI) | 0.73 (0.629, 0.818) | 0.82 (0.735, 0.898) | 0.806 (0.71, 0.881) |
| AUC | 0.743 | 0.88 | 0.87 |
| Kappa | 0.45 | 0.629 | 0.573 |
| Sensitivity | 0.71 | 0.812 | 0.793 |
| Specificity | 0.74 | 0.836 | 0.812 |
| Positive predicted value | 0.658 | 0.72 | 0.657 |
| Negative predicted value | 0.788 | 0.89 | 0.896 |

performed LOOCV performance evaluation and prediction threshold calibration balancing prediction sensitivity and specificity (*Figure 5*). The model predicting AIS improvement had a cross-validated AUC of 0.74, the model predicting AIS A at discharge had a cross-validated AUC of 0.88, and the model predicting AIS D at discharge had a cross-validation AUC of 0.84. Other metrics of classification performance can be seen in *Table 12*. Both the parsimonious model predicting AIS improvement and the one predicting AIS A at discharge included quadratic aMAP as an important predictor. The model predicting AIS A also included the time of MAP out of range 76–117 mmHg. The model predicting AIS D did not include any of the MAP associated terms, suggesting that patients discharged with AIS D can be predicted without considering their MAP during OR. In fact, training the same model but with the inclusion of the quadratic aMAP term resulted in slightly worse prediction performance (AUC 0.84 vs. 0.83). Training the models predicting AIS improvement and AIS A at discharge but without a



**Figure 5.** Leave-one-out cross-validation (LOOCV) performance of prediction models. We built three prediction models, one to predict American Spinal Injury Association (ASIA) Impairment Scale (AIS) improvement of at least one grade at discharge (AIS impro., **a**), one to predict AIS A at discharge (A at disch., **b**) and one to predict AIS D at discharge (D at disch., **c**). The sensitivity and specificity for each model was computed out of the prediction probability of LOOCV, where each leave-one-out patient is predicted with the model that was trained without that patient. For each model, the classification threshold was set at the probability that balances sensitivity and specificity (dashed red line). The receiving operation curve (ROC) and area under the curve (AUC) for the three models are presented in **d**.

MAP component (quadratic MAP term or time of MAP out of range) reduced the model performance considerably (AUC, AIS improvement: 0.74 vs. 0.52; AIS A discharge: 0.88 vs. 0.78).

Altogether, this suggests that models can be built for predicting AIS improvement or AIS A at discharge and that such the model performance critically depends on MAP during OR. Conversely, we did not find evidence that predicting AIS D at hospital discharge is dependent on intra-operative MAP.

## Discussion

Acute hypotension is common in patients with SCI due to neurogenic shock (*Lehmann et al., 1987*; *Krassioukov et al., 2007*) and autonomic dysregulation (*Lehmann et al., 1987*), probably contributing to post-traumatic spinal ischemia (*Streijger et al., 2018*; *Hall and Wolf, 1987*), which is known

to cause impaired neurological recovery in animal models (*Fehlings et al., 1989*). Level 4 evidence from a small single-center case series study in the 1990s suggested that MAP augmentation to 85–90 mmHg during the first 5–7 days after injury was linked to neurological recovery in acute SCI (*Levi et al., 1993*; *Vale et al., 1997*). These results are the basis of clinical guidelines for avoidance of hypotension in acute SCI management (*Aarabi et al., 2013*). However, while numerous clinical studies support MAP augmentation, the arbitrary, recommended MAP goal has been controversial (*Cohn et al., 2010*; *Hawryluk et al., 2015*; *Saadeh et al., 2017*). Recent analysis of high-frequency ICU monitoring data (*Hawryluk et al., 2015*) and systematic meta-analysis of post-surgery management (*Saadeh et al., 2017*) suggest that the MAP threshold to avoid is actually lower (~75 mmHg) than the current recommendation of 85 mmHg, and that MAP management might be effective at shorter duration (< 5 days post-injury) than the 7 -day goal (*Saadeh et al., 2017*). The present study represents a multicenter, data-driven, and cross-validated re-evaluation in a different setting (during surgery as compared with prior ICU studies).

Our analysis support that there must be a MAP range during surgery at which neurological recovery is maximized, providing further evidence that MAP management for maintaining normotension might be more beneficial for patient outcome than MAP augmentation for hypotension avoidance alone (*Ehsanian, 2020*; *Nielson et al., 2015*). The low boundary of 76 mmHg found in our ultra-early analysis further supports previous suggestions for lowering the intervention threshold (*Cohn et al., 2010*; *Hawryluk et al., 2015*; *Saadeh et al., 2017*). On the other side, we find an upper boundary to MAP management between 104 and 117 mmHg, above which the probability of improvement is reduced. Thus, the proposal for MAP augmentation with vasopressors to increase spinal cord perfusion (*Saadoun and Papadopoulos, 2016*) has a limit since spinal hyper-perfusion pressure can be detrimental (*Saadoun and Papadopoulos, 2016*). The physiological rational is that high blood pressure induced by vasopressors can translate to increased risk of hemorrhage in the injured spinal cord, exacerbating tissue damage (*Soubeyrand et al., 2014*; *Streijger et al., 2018*; *Guha et al., 1987*). Moreover, the use of some vasopressors might cause more complications in patients (*Inoue et al., 2014*) while also potentially contributing to intra-spinal hemorrhage. In fact, recent results in acute experimental SCI suggest controlling for hemodynamic dysregulation through a cardiac-focused treatment instead of using standard vasopressors such as norepinephrine (*Williams et al., 2020*). Specifically, the authors demonstrated that dobutamine can correct for hemodynamic anomalies and increase blood flow to the spinal cord while reducing the risk of hemorrhage compared to norepinephrine. Furthermore, hypertension during surgery in rodent SCI has been associated with lower probability of recovery (*Nielson et al., 2015*), probably related to higher tissue damage. Our findings together with previous work (*Ehsanian, 2020*) also translate these animal study results to humans, indicating that prolonged periods of hypertension early after injury can be a predictor of poor neurological recovery in patients with SCI.

An important finding of our study is the indication that level of injury and injury severity modify the association of MAP with neurological recovery. We observed that normotensive MAP during surgery predicts AIS improvement in patients with cervical SCI but not in patients with lower injuries (thoracic, lumbar, and sacral). While the heterogeneity of our population and low sample size for patients with non-cervical SCI sets limitations on interpreting the results, our finding raises a relevant question regarding precision management of patients with SCI. Patients with cervical SCI present more frequently with hemodynamic and cardiac abnormalities than patients with thoracolumbar SCI, increasing the need for treatment (*Lehmann et al., 1987*). This is due to sympathetic dysregulation in upper cord injuries, which reduces sympathetic tone likely causing reduced heart contractility, bradycardia, and hypotension (*Lehmann et al., 1987*; *Myers et al., 2007*; *Teasell et al., 2000*). This is particularly true for individuals with severe cervical injury (*Lehmann et al., 1987*). In that context, our results may indicate that those patients with cervical SCI that are more difficult to maintain within a normotensive MAP are probably less likely to improve in neurological function. Alternatively, it could also be the case that more aggressive MAP management treatment is performed in these patients during their course in the hospital, which could increase the chances of aggravating secondary cord injury. Hemodynamic instability early after injury could serve as a prognostic physiology-based biomarker in a subset of the population, providing a potential tool for precision medicine in SCI. Hence, we have established basic prediction models around non-linear features of MAP that could serve as a benchmark for future machine learning development.

Another relevant contribution of this work is the analytical workflow. First, we demonstrate that topology-based analytics can undercover associations for hypothesis generation during exploratory analysis in a cross-species validation. Our group has previously used a similar workflow in data from animal models (*Nielson et al., 2015*) suggesting that hypertension is a predictor of neurological recovery and providing rational for the present study. Hence, our work constitutes a successful story of translating machine intelligence analytical tools from animals to humans. Second, we provide further illustration that patient similarity networks are useful and interpretable representations of multidimensional datasets that capture associations during exploratory analysis that can then be validated by network-independent confirmatory analysis. Third, we successfully combine Isomap, a non-linear dimensionality reduction method, with topology-based metrics to evaluate embedding solutions. Fourth, our method for finding the MAP range could be expanded and deployed in other settings. Lastly, our workflow captures representations of multidimensional time-series of different lengths into a network that is actionable.

Limitations of this study include the retrospective nature of the analysis, the relatively small sample size (although large for SCI), and the use of an estimated ordinal scale (AIS grade) as an indicator of neurological recovery. An important consideration is the difficulty of determining AIS grade early after injury. Moreover, other factors not considered in this analysis such as MAP levels before or after surgery or the use of vasopressors might influence the results. Future research with more granular data should address these and other important questions.

## Acknowledgements

Supported in part by NIH/NINDS: R01NS088475 (ARF); R01NS122888 (ARF); UH3NS106899 (ARF); U24NS122732 (ARF); Department of Veterans Affairs: 1I01RX002245 (ARF), I01RX002787 (ARF); Wings for Life Foundation (ATE, ARF); Craig H Neilsen Foundation (ARF); DOD: SC150198 (MSB); SC190233 (MSB); DOE: DE-AC02-05CH11231 (DM).

## Additional information

### Group author details

**The TRACK-SCI Investigators**
MS Beattie; JC Bresnahan; JF Burke; A Chou; CA de Almeida; SS Dhall; AM DiGiorgio; X Doung-Fernandez; AR Ferguson; J Haefeli; DD Hemmerle; JR Huie; N Kyritsis; GT Manley; S Moncivais; C Omondi; JZ Pan; LU Pascual; V Singh; JF Talbott; LH Thomas; A Torres-Espin; P Weinstein; WD Whetstone

### Funding

| Funder | Grant reference number | Author |
|---|---|---|
| National Institute of Neurological Disorders and Stroke | R01NS088475 | Adam R Ferguson |
| National Institute of Neurological Disorders and Stroke | R01NS122888 | Adam R Ferguson |
| National Institute of Neurological Disorders and Stroke | UH3NS106899 | Adam R Ferguson |
| U.S. Department of Veterans Affairs | 1I01RX002245 | Adam R Ferguson |
| U.S. Department of Veterans Affairs | I01RX002787 | Adam R Ferguson |

| Funder | Grant reference number | Author |
| --- | --- | --- |
| Wings for Life | | Abel Torres-Espín<br>Adam R Ferguson |
| Craig H. Neilsen Foundation | | Adam R Ferguson |
| Department of Defense | SC150198 | Michael S Beattie |
| Department of Defense | SC190233 | Michael S Beattie |
| Foundation for Anesthesia Education and Research | A123320 | Jonathan Z Pan |
| Department of Energy | DE-AC02-05CH11231 | Dmitriy Morozov |
| National Institute of Neurological Disorders and Stroke | U24NS122732 | Adam R Ferguson |

The funders had no role in study design, data collection and interpretation, or the decision to submit the work for publication.

## Author contributions

Abel Torres-Espín, Conceptualization, Formal analysis, Investigation, Methodology, Software, Validation, Visualization, Writing – original draft, Writing – review and editing; Jenny Haefeli, Conceptualization, Formal analysis, Investigation, Methodology, Writing – review and editing; Reza Ehsanian, Writing – review and editing, Data interpretation; Dolores Torres, Data curation, Writing – review and editing, Data curation; Carlos A Almeida, Benjamin Dirlikov, Debra D Hemmerle, Data curation, Writing – review and editing; J Russell Huie, Writing – review and editing, Data interpretation; Austin Chou, Writing – review and editing, Data interpretation; Dmitriy Morozov, Formal analysis, Methodology, Visualization, Writing – review and editing, Data interpretation; Nicole Sanderson, Formal analysis, Methodology, Visualization, Writing – review and editing, Data interpretation; Catherine G Suen, Data curation, Writing – review and editing, Data interpretation; Jessica L Nielson, Writing – review and editing, Data interpretation; Nikos Kyritsis, Writing – review and editing, Data interpretation; Jason F Talbott, Conceptualization, Investigation, Methodology, Writing – review and editing; Geoffrey T Manley, Conceptualization, Resources, Writing – review and editing; Sanjay S Dhall, Conceptualization, Resources, Writing – review and editing; William D Whetstone, Data curation, Methodology, Validation, Writing – review and editing; Jacqueline C Bresnahan, Conceptualization, Resources, Writing – review and editing; Michael S Beattie, Conceptualization, Resources, Writing – review and editing, Data interpretation; Stephen L McKenna, Conceptualization, Data curation, Methodology, Resources, Writing – review and editing, Data interpretation; Jonathan Z Pan, Conceptualization, Data curation, Methodology, Writing – review and editing, Data interpretation; Adam R Ferguson, Conceptualization, Formal analysis, Funding acquisition, Methodology, Supervision, Writing – original draft, Writing – review and editing, Data interpretation

## Author ORCIDs

Abel Torres-Espín http://orcid.org/0000-0002-9787-8738
Nikos Kyritsis http://orcid.org/0000-0001-7801-5796
Debra D Hemmerle http://orcid.org/0000-0003-2796-6107
Jonathan Z Pan http://orcid.org/0000-0001-5814-3707
Adam R Ferguson http://orcid.org/0000-0001-7102-1608

## Ethics

Human subjects: This study constitutes a retrospective data analysis. All data was de-identified before pre-processing and analysis. Protocols for retrospective data extraction were approved by Institutional Research Board (IRB) under protocol numbers 11-07639 and 11-06997.

## Decision letter and Author response

Decision letter https://doi.org/10.7554/eLife.68015.sa1
Author response https://doi.org/10.7554/eLife.68015.sa2

## Additional files

### Supplementary files
• Supplementary file 1. Code.output.html: output html file results of running the source code.

• Source code 1. Source.code.Rmd: Rmarkdown file containing the source code that reproduced the paper.

• Transparent reporting form

### Data availability
Source data has been deposited to the Open Data Commons for Spinal Cord Injury (odc-sci.org; RRID:SCR_016673) under the accession number ODC-SCI:245 (https://doi.org/10.34945/F5R59J) and ODC-SCI:246 (https://doi.org/10.34945/F5MG68).

The following dataset was generated:

| Author(s) | Year | Dataset title | Dataset URL | Database and Identifier |
|---|---|---|---|---|
| Torres-Espin A, Haefeli J, Ehsanian R, Torres D, Almeida C, Huie J, Chou A, Dirlikov B, Suen C G, Nielson JL, Kyritsis N, Duong-Fernandez X, Thomas LH, Hemmerle T, Morozov DD, Sanderson D, Talbott N, Manley J, Dhall GT, Whetstone SS, Bresnahan WD, Beattie JC, McKenna MS, Pan SL, Ferguson JZ | 2021 | ASIA Impairment Scale, level of injury, intraoperative time series mean arterial pressure and heart rate after spinal cord injury in patients in a multi-site retrospective TRACK-SCI cohort: site 1 of 2 | https://doi.org/10.34945/F5R59J | Open Data Commons for Spinal Cord Injury, 10.34945/F5R59J |
| Torres-Espín A, Haefeli J, Ehsanian R, Torres D, Almeida C A, Huie J, Chou A, Dirlikov B, Suen CG, Nielson JL, Kyritsis N, Duong-Fernandez X, Thomas LH, Hemmerle DD, Morozov D, Sanderson N, Talbott JF, Manley GT, Dhall SS, Whetstone WD, Bresnahan J C, Beattie MS, McKenna SL, Pan JZ, Ferguson AR | 2021 | Intraoperative time series mean arterial pressure and heart rate after spinal cord injury in patients in a multi-site retrospective TRACK-SCI cohort: site 2 of 2 | https://doi.org/10.34945/F5MG68 | Open Data Commons for Spinal Cord Injury, 10.34945/F5MG68 |

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
