## [Editor Report]

The major strengths of this paper are the use of a combination of relatively novel approaches/applications to the identification of important predictors for recovery after spinal cord surgery. These include various data reduction techniques such as dissimilarity matrices and a subject-centered topological network analysis to identify phenotypes.

---

## [Decision Letter]

**Decision letter after peer review:**

Thank you for submitting your article "Topological network analysis of patient similarity for precision management of acute blood pressure in spinal cord injury" for consideration by *eLife*. Your article has been reviewed by 3 peer reviewers, and the evaluation has been overseen by a Reviewing Editor and a Senior Editor. The following individual involved in review of your submission has agreed to reveal their identity: Marcel Kopp (Reviewer #2).

The Reviewers and Editors have discussed the reviews with one another, and this decision letter is to help you prepare a revised submission.

Essential revisions:

1) Please provide information regarding what covariate adjustment was used in the confirmatory logistic regression models.

2) Can the authors provide an explanation of why they chose the specific forms of clustering to identify patient phenotypes? Other, perhaps simpler and more common unsupervised machine learning algorithms could have been used.

3) Are the results sensitive to the defined outcome of improvement of at least one AIS grade? What happens if this is increased to e.g. 2 grades?

4) Two different approaches to analysis were used – i.e. essentially clustering of some form and also logistic regression (using e.g. quadratic and spline functions). Can the authors comment on whether these 2 approaches can be used interchangeably or whether one would be preferred over to the other to answer the research questions of interest. What advantage does the clustering approach have in terms of the research question?

5) Why was a simple accuracy metric used and not e.g. AUC? Does the accuracy metric account for an imbalance in the outcomes?

6) The LOOCV accuracy was not that high suggesting a lot of other factors might influence outcomes. Is the accuracy really high enough to support the use of MAP being used by clinicians for decision making and/or interventions to control MAP during surgery?

7) What variables were used for the LASSO. The prediction accuracy again seems very low for high-dimensional dataset.

8) Only one dataset was used without splitting the data into a training and validation dataset. Are similar results for the topological network analysis obtained if the data is split for training and validation?

9) What was the modularity of the final network and does it suggest significant clustering?

10) Why was days from surgery to discharge used in the logistic regression models? Might it not be considered a mediator rather than a confounder – and how does its exclusion from the model influence the result?

11) The limitations are mentioned but not discussed or justified. This leads to the following questions: a) Why was the lesion level not included in the analysis? and b) Why did the authors only analyze MAD values during surgery? Because the analysis of MAP data from the ICU period published elsewhere showed similar results regarding the lower limit of MAP, wouldn't it be of interest to know how much overlap there is between the populations with critical MAP values during surgery and during stay in the ICU?

12) Introduction: Neither hypertension nor hypotension following acute SCI has been conclusively demonstrated to impact neurological recovery. Instead, guidelines and more recent work are based purely on observations and post-hoc regression analyses. While the purported mechanism for repeated hypotensive episodes is clear, readers may benefit from at least a brief description of why both hypertension and hypotension could plausibly be important (aside from the fact that, again, non-causal observations demonstrated a relationship in the author's prior work).

13) AIS scores: Based on Table 1 most patients were discharged within 2 weeks after injury. The neurological exam is not so reliable at this point. This is a big limitation of the current work. Although a six month follow-up would be ideal to determine whether neurological recovery occurred, the authors should at least mention this.

14) All the analyses seem to have been conducted on an AIS change. The authors should demonstrate that their analysis holds for a more linear measure of recovery (e.g., total motor score).

15) Based on Figure 2 – Supplement 2, it is difficult to ascertain whether clusters contain a higher proportion of individuals that show an AIS improvement, and those individuals tend to have a MAP > 80 and < 100, is due to these clusters having individuals who were less severe to begin with (i.e., C,D,E) and therefore less likely to be hemodynamically unstable. One way to answer this would be to examine AIS A patients in a separate analysis and determine whether these findings hold. Because, the alternative explanation here is that this analysis is effectively finding a proxy for initial injury severity (i.e., more severe, more hemodynamically unstable) – and not that hemodynamic instability per se is the problem. Another analysis that could help complement this work and avoid this confound would be to use total motor score as the outcome instead of AIS conversion.

16) Logistic regression – Based on Figure 2p the overall trend looks more to be that higher MAP = > Pr(δ AIS grade > 0). The exception is only 2 data points on the top end. It is difficult to determine how robust the notion is that there is a 'too high' component to this data. Indeed it seems that a linear model does quite well for this analysis as well. Please see my comment below but this should be addressed as the concept of also having a 'top cutoff' is an extremely important clinical feature here.

17) Time outside MAP – The authors use an approach that systematically increases their window in both directions to find their optimal range of 76-104. However, what happens if you then hold 76 and only increase on the upper end? Does this rapidly degrade the relationship? If not, again this would suggest that the evidence for a top end cut-off is not as strong. While I understand the authors briefly looked at this (methods) it seems worth exploring further as this is a critical point for clinical management. I do not see a good reason that the time outside the threshold can not at least be plotted to determine this relationship.

18) Data availability – The code and analysis should be made available to the reviewers. It is impossible to determine the accuracy of this type of analysis without it.

19) Discussion – It seems that the authors should discuss the confound of injury severity being linked with worse hemodynamic instability, and also worse neurological recovery. It would be helpful to include some of the suggested analyses to convince the readers that this confound does not explain the results since it is the most likely alternative explanation.

*Reviewer #1:*

The major strengths of this paper are the use of a combination of relatively novel approaches/applications to the identification of important predictors for recovery after spinal cord surgery. These include various data reduction techniques such as dissimilarity matrices and a subject-centred topological network analysis to identify phenotypes. The weaknesses include its relatively modest prediction accuracy and the lack of internal and external validation in the primary network analysis.

*Reviewer #2:*

The major strength of the paper is the statistically highly advanced analysis based on high-resolution data from acute SCI care, i.e. intra-operative mean arterial blood pressure (MAP) and heart rate. The steps of data exploration and analysis and their results are presented transparently. In conjunction with the results of previous studies suggesting that the lower threshold of MAP levels to be avoided in the ICU is about 75 mmHg, the main results of the this study imply that the minimum target MAP may be lower than the currently recommended 85 mmHg also during surgery. The analysis, which combines machine learning algorithms with logistic regression models, may serve as a template for data-driven studies also on other aspects of critical care in the field of SCI.

A weakness of the study is that some of the baseline neurological criteria were not included in the analysis. In particular, the neurological level of injury could be important for the research question, because the degree of blood pressure dysregulation also depends on the lesion level. The authors mention this limitation but do not explain why they accept it. Another limitation is the relatively small sample size of the study. Therefore, the specific results might have limited generalizability. Nevertheless, the study is an essential contribution to the readjustment of MAP threshold recommendations in the very acute stage of SCI and provides key information for the design of future precision medicine studies.

*Reviewer #3:*

The authors present a nice analysis of the relationship between intraoperative mean arterial pressure and neurological recovery after spinal cord injury, and I appreciate the opportunity to review this work. In general, the results are interesting and largely in line with a growing body of evidence that supports the use of hemodynamic monitoring in the acute phase after injury. There are three primary points with regard to this work that the authors can and should address prior to publication.

First, the exclusive use of AIS conversion as the outcome of neurological improvement is not ideal. Demonstrating that these findings are robust to other more continuous measures of neurological improvement such as motor/sensory scores would go a long way towards demonstrating that this finding is robust.

Second, the authors state in the discussion that MAP management may only be needed for <5 days post-injury. There does not appear to be strong data in the paper to support this point. Either more data should be added that supports this contention or this point should be removed.

Third, although the authors briefly discuss the confound of injury severity being linked to hemodynamic instability, the results are compelling enough at the moment to discount a role of injury severity. Individuals with more severe injuries will necessarily have greater hemodynamic instability, particularly in the hyperacute and acute phase after injury. These same individuals are also less likely to exhibit a conversion of their AIS score (e.g., individuals with AIS A/B). The authors control for this in some of their analyses (e.g., including a coefficient of initial AIS score in their regression models), yet their results seem to indicate that the clusters of individuals they focus on are indeed those with less severe injuries to start with. Demonstrating that injury severity is not a factor would require an analysis of only individuals with AIS A injuries at admission. This would be very compelling and enhance the impact of this work.

Overall, this is an interesting analysis that will be of use to the field.

---

## [Author Response]

Essential revisions:1) Please provide information regarding what covariate adjustment was used in the confirmatory logistic regression models.

We adjusted the models by heart rate, length of surgery, days from surgery to discharge, age, and AIS grade at admission. We added text to the Results section that helps to clarify the models and the covariates.

2) Can the authors provide an explanation of why they chose the specific forms of clustering to identify patient phenotypes? Other, perhaps simpler and more common unsupervised machine learning algorithms could have been used.

We have added further explanation to the manuscript. In short, we and others have previously shown that topological analysis drives discovery of biologically meaningful associations with high resolution (Nielson et al., 2015; Nicolau et al., 2011). Here, we use topological network analysis for several reasons. First, physiological data is non-linear, which conforms to a manifold in the multidimensional space. We use Isomap, a well-established non-linear dimension reduction method, to find the low dimensional embedding of the data (Balasubramanian, 2002; Tenenbaum, 2000). Isomap works by constructing a neighbor network to approximate the geodesic distance between observations. Second, to ensure that Isomap approximates the shape of the manifold we use topological metrics. Third, the optimal Isomap solution had 4 dimensions, which makes it challenging for representing data for exploration. We construct the network of patients as networks are efficient data representation methods for multidimensional data. Therefore, using the walktrap clustering algorithm is a natural progression as it is based on the network and its topology.

Indeed, other methods could be used, however, it has been shown that networks (graphs) allow for more efficient data exploration than other approaches as they are capable of representing multidimensional relationships on a single 2D visual space (Benson et al., 2016). Ultimately, the question of which unsupervised method to use in each case is a scholarly question that requires its own dedicated research. Here, we focused on topological methods as we have successfully used them before to explore hemodynamics after SCI in animal models, a direct bench-to-bedside translation of both physiological variables and analytical methods (Nielson et al., 2015).

3) Are the results sensitive to the defined outcome of improvement of at least one AIS grade? What happens if this is increased to e.g. 2 grades?

This is a good question. Unfortunately, we do not have enough data to provide an accurate answer, despite having one of the largest cohorts assembled in SCI (N = 118, greater than 80% of acute SCI studies registered in Clinical trials.gov). Though increasing one AIS grade is clinically meaningful, we acknowledge the limitations of our study and agree with the reviewers that an analysis using a more sensitive outcome such as a continuous metric (e.g. motor/sensory scores) would increase the value of our findings. This is an area for future research that will benefit from the ongoing assembly of ever larger cohorts in the field.

4) Two different approaches to analysis were used – i.e. essentially clustering of some form and also logistic regression (using e.g. quadratic and spline functions). Can the authors comment on whether these 2 approaches can be used interchangeably or whether one would be preferred over to the other to answer the research questions of interest. What advantage does the clustering approach have in terms of the research question?

We have now clarified this in the results and discussion. These two methods are complementary rather than interchangeable. The patient network analysis is unsupervised and was used with the intention of exploratory discovery. It is through this method where we observed the potential double bound (upper and lower limits) for arterial pressure in humans, and the dependence of it to AIS at admission. The logistic regression is used in two ways, one to confirm the hypothesis formulated through the network analysis and two to find a precise MAP threshold through LASSO regularization. The fact that the MAP threshold for prediction of improvement captured through the network is very close to the one found by the LASSO indicates that our network methodology captures valid relationships in the data without previous knowledge of outcome (unsupervised). This suggests that our network method can be used for data discovery in future research including more physiological measures. We take the values obtained through LASSO as more confirmatory because it is a modeling approach, while the network analysis is exploratory. We added text in the manuscript clarifying these points

The use of topological analysis and clustering through networks has some advantages for exploratory analysis and discovery in the question at hand. We show that even though we are technically only including 2 variables in the analysis (MAP and HR), the time series of these two conform to a multidimensional manifold that we approximate by Isomap plus topological metrics. The network allows us to represent that multidimensional space in a 2D visual based on the similarity between patients. This is convenient for fast exploration of the multidimensional space, where relationship between patient similarities and any variable can be easily depicted by mapping the variable values to coloring the nodes. This and other related topological data analysis methods have been described as “Machine Intelligence” approaches as they aid the analyst with expert matter knowledge to perform fast discovery (Carlsson, 2009; Wasserman, 2018).

5) Why was a simple accuracy metric used and not e.g. AUC? Does the accuracy metric account for an imbalance in the outcomes?

We have now performed new analysis on prediction modeling. For these, we have calculated AUC and ROC. We have included new text and a new figure reflecting the new results.

6) The LOOCV accuracy was not that high suggesting a lot of other factors might influence outcomes. Is the accuracy really high enough to support the use of MAP being used by clinicians for decision making and/or interventions to control MAP during surgery?

We have included new analysis for the prediction modeling and added a subsection in the results. In these, we show that the prediction accuracy can be improved through model selection, where MAP is an important feature to maintain in the model for good prediction performance, especially for predicting AIS improvement or AIS A at discharge. It is important to note that all our metrics of performance are for LOOCV, which measures performances over unseen data for n (patients) trained models and thus offers an estimation of the generatability of the data and some protection against model overfitting. In that sense, LOOCV accuracy would be in most cases lower than accuracy of a model that has not been measured with cross-validation, which is the case of most logistic regression in the clinical literature. With the new prediction modeling we included, our model has a LOOCV AUC of 0.75 for predicting AIS improvement at discharge. While we acknowledge the limitations of the study, we think that our prediction models are informative, and indeed are among the first ever to rigorously test ultra-acute predictors of outcome in SCI. These models can provide a benchmark for future prediction modeling in the early level-1 trauma clinical SCI.

7) What variables were used for the LASSO. The prediction accuracy again seems very low for high-dimensional dataset.

We have added the range threshold analysis in its own section in the results and separated the explanation in the methods to address this point. LASSO was used to determine the most predictive MAP range and respective thresholds. The variables included are the time spent out of each one of the MAP ranges. In that sense, we have not used LASSO as supervised predictive ML method but rather as a feature selection method (through the regularization process) to obtain the last standing coefficient and correspondingly the best predicting MAP range. Therefore, we do not expect our use of LASSO to produce the most accurate predictive model but to provide the most predictive MAP range.

8) Only one dataset was used without splitting the data into a training and validation dataset. Are similar results for the topological network analysis obtained if the data is split for training and validation?

We agree with the reviewer about the need to validate the results and have added explicit discussion of this point. Topological network analysis learns the cross-validated data manifold space, through a specific type of validation that subsamples the subspace of all networks. Splitting between validation and training in the exploratory topological analysis presents two major challenges: (1) splitting the data with our sample size can be problematic since would leave relatively few cases for training and testing with precision; (2) training and testing split is normally performed in supervised models to determine how well a trained model can predict unseen data. The topological network analysis process that we implemented is an unsupervised method used to explore the relationship between patients and physiological variables. In that sense, we are not using the network as a model to be “trained” but as a descriptive statistic of the shape of the data manifold. Nonetheless, the metrics of topological descriptions that we use (persistent homology) find the best configuration (most stable) given the data's many potential configurations (Carlsson, 2009). In that sense, we are validating the topological structure of the network for our specific case.

We conducted further validation by (1) performing confirmatory analysis through hypothesis testing (logistic models) and (2) using leave-one-out cross-validation (LOOCV). LOOCV is perhaps the best alternative since splitting the data in this case would leave very few cases for training and testing which would ultimately underpower the analysis.

9) What was the modularity of the final network and does it suggest significant clustering?

The modularity of the final network was Q = 0.837. Modularity has a maximum of 1, representing complete clustering above expected in a random graph with the same structure (same number of nodes and degrees). Our final Q is quite high, suggesting the presence of clusters above what should be expected by chance. We have included this information in the corresponding figure legend.

10) Why was days from surgery to discharge used in the logistic regression models? Might it not be considered a mediator rather than a confounder – and how does its exclusion from the model influence the result?

We now include discussion of this important point.

We use time of surgery to discharge in the model as a covariate, to control for the potential effect in outcome related to the time to discharge. Considering time of surgery to discharge as a mediator is indeed a possibility. That would be true if: (a) time to discharge affects probability of improving (the outcome), and (b) MAP dysregulation during surgery affects days from surgery to discharge. The logical progression in the causal pathway would be as follows: higher MAP OR dysregulation indicates/causes changes in time to discharge, and time to discharge then affects probability of improvement (either increase or decrease). Since we see that MAP dysregulation (or proxies of it) actually reduces probability of improvement, then there are two options: (1) MAP dysregulation during OR directly or indirectly associates with longer time to discharge, and higher time to discharge reduce probability of improvement, or (2) MAP dysregulation associates with shorter time to discharge which in turns, is related to reduced probability of improvement. One would expect that the longer the patient is in the hospital, the more time the patient has to improve before discharge. However, the longer a patient stays in the hospital is probably an indicator of their severity. We have investigated this matter using different models to dissect the potential causal pathway.

First, we fitted a logistic model with AIS improvement as response and days to discharge as the only predictor. This resulted in a non-significant p value of p = 0.32, suggesting that days to discharge doesn’t associate with probability of improvement. Second, we fitted a linear model with days to discharge as a response and quadratic aMAP (both linear and quadratic terms) as predictors. This resulted as a significant coefficient of the quadratic term (p = 0.047), although the model was not significant (p = 0.13 for the F statistic) and the adjusted R^2^ was small (0.0217). Visual examination of the data indicates that the density of patients that tend to stay longer in the hospital are the ones with a MAP in the normotensive range ~ 80-100. We also investigated whether days to discharge interacts with MAP and quadratic MAP to predict AIS improvement, with no significant results on the interaction, and the inclusion of the interaction do not change the quadratic association of MAP with the response. These suggest that these two factors do not moderate each other. We also performed the same analysis with time MAP out of 76-104 mmHg instead of aMAP as a different proxy for MAP dysregulation, with similar results.

Finally, eliminating days to discharge from the full covariate model predicting AIS improvement doesn’t have a major effect on the model fit. A likelihood ratio test between both models shows a non-significant change in variance explained (p=0.79) with a deviance difference of ~ 0.1%.

Therefore, given the data, we think there is little evidence for considering an effect of MAP dysregulation on AIS improvement mediated (or moderated) by length of stay in the hospital. We have maintained days from surgery to discharge as covariate because although not significant, we still think it is important to adjust the model for differences in length of hospital stay between patients. We have included this analysis in the manuscript as results. In addition, we understand the limitation of the above analysis where all other variables and their relationship have not been considered. Nonetheless, we want to acknowledge the importance of dissecting the correlational observations as a first approximation to determine the causal effects. We will definitely explore these ideas in the near future as larger cohorts of data become available.

11) The limitations are mentioned but not discussed or justified. This leads to the following questions: a) Why was the lesion level not included in the analysis? and b) Why did the authors only analyze MAD values during surgery? Because the analysis of MAP data from the ICU period published elsewhere showed similar results regarding the lower limit of MAP, wouldn't it be of interest to know how much overlap there is between the populations with critical MAP values during surgery and during stay in the ICU?

We thank the reviewers for pointing to these important questions. We have expanded our analysis to include the level of injury. We found that the association of MAP range of AIS grade improvement is dependent on whether the patient had a cervical injury or not. At this time, we do not have enough samples to tackle this question with higher granularity (e.g. per segment level) as our data include very few cases with middle and lower level injuries. This is a limitation due to the infrequent nature of SCI, the difference in injury incidence per segment level, and the slow accumulation of patient cohorts, even at some of the busiest level 1 trauma centers in the country. We have included a new section in the results and expanded the discussion accordingly.

Regarding the analysis of MAP only during surgery, there are some considerations: First, our previous results in animal models suggest that OR hemodynamics is an important predictor of recovery. Second, this is a retrospective cohort in which systematic ICU monitoring is not available for answering the proposed question. Third, and most importantly, monitoring in the OR is perhaps the best controlled scenario for hemodynamic monitoring in these patients. We acknowledge the importance of the question of whether MAP during OR is sufficient as a proxy for patient dysregulation and how it confounds with hemodynamic dysregulation/treatment during ICU. As part of the TRACK-SCI study, we are collecting high-frequency monitoring data in a prospective cohort and we will explore these questions in the future.

12) Introduction: Neither hypertension nor hypotension following acute SCI has been conclusively demonstrated to impact neurological recovery. Instead, guidelines and more recent work are based purely on observations and post-hoc regression analyses. While the purported mechanism for repeated hypotensive episodes is clear, readers may benefit from at least a brief description of why both hypertension and hypotension could plausibly be important (aside from the fact that, again, non-causal observations demonstrated a relationship in the author's prior work).

We have included further details in the introduction describing the importance of both hypertension and hypotension from clinical and animal studies. We agree with the reviewers that a causal link between hemodynamic control and recovery in humans remains elusive, and it should be noted that randomized controlled studies are challenging in this context given the ethical concerns of such studies. Nevertheless, some experimental animal studies 'clamping MAP' at different set points strongly suggest causal effects (Nout et al., 2012), and physician-scientists in the field are conducting ongoing studies that appear strongly supports a potential causal role. Such a causal role is theoretically well-founded, given that in traumatic brain injury and stroke, tissue perfusion/oxygenation, and brain tissue survival are supported by prevention of hypotension, but it is also possible that hypertension can induce 'hemorrhagic conversion' and bleeding into the tissue with devastating effects on lesion expansion. Spinal cord injury is known to involve similar secondary injury processes in both rodent and primate models (Crowe et al., 1997).

13) AIS scores: Based on Table 1 most patients were discharged within 2 weeks after injury. The neurological exam is not so reliable at this point. This is a big limitation of the current work. Although a six month follow-up would be ideal to determine whether neurological recovery occurred, the authors should at least mention this.

We agree with the limitations and have added some text in the discussion. These data are unique in that they reflect ultra-acute data from the level 1 trauma center. This is a moment in the care pathway when a highly heterogeneous patient population is concentrated in one location prior to being dispersed to a broad variety of discharge scenarios (largely depending on insurance status in the US healthcare system). This makes long term follow-up data difficult to obtain. Nevertheless, as the TRACK-SCI prospective study progresses, we are collecting contact information and have funds to track patients out to 12 months follow up together with high granular physiological data both in the OR and ICU. It will take many years to collect a large N, but in the future this dataset will provide a rich source of information to continue researching these important questions.

14) All the analyses seem to have been conducted on an AIS change. The authors should demonstrate that their analysis holds for a more linear measure of recovery (e.g., total motor score).

We agree with the reviewers that AIS change has limitations. We now acknowledge them in the manuscript. We have also added further analysis as a result of this revision. Specifically, we investigated models predicting AIS A and AIS D at discharge in addition to the AIS change analysis we had. Please see the following responses to details on the new analysis.

15) Based on Figure 2 – Supplement 2, it is difficult to ascertain whether clusters contain a higher proportion of individuals that show an AIS improvement, and those individuals tend to have a MAP > 80 and < 100, is due to these clusters having individuals who were less severe to begin with (i.e., C,D,E) and therefore less likely to be hemodynamically unstable. One way to answer this would be to examine AIS A patients in a separate analysis and determine whether these findings hold. Because, the alternative explanation here is that this analysis is effectively finding a proxy for initial injury severity (i.e., more severe, more hemodynamically unstable) – and not that hemodynamic instability per se is the problem. Another analysis that could help complement this work and avoid this confound would be to use total motor score as the outcome instead of AIS conversion.

This is an important consideration. We study the question of injury severity being a driver for the effect in several analyses. First, as suggested by the reviewers, we conducted the model fitting only in patients with AIS A at admission. This resulted in a significant linear MAP term, and a quadratic MAP with the second biggest coefficient and a p = 0.14. Comparing a linear model with the quadratic one in this cohort through a likelihood ratio test resulted in a p = 0.07. This suggests that while the linear term of MAP as predictor of AIS improvement is maintained significantly for AIS A patients, the evidence for a quadratic form is weak in this cohort. However, with the reduction in sample size (n=46) we may be underpowered for detecting such an effect. Second, we have included AIS at admission as a covariate in the full model to adjust for potential differences between different injury severities. In this scenario, the quadratic form of MAP predicting AIS improvement is maintained. All together, this suggests that if non-linear relationship of MAP and AIS improvement depend on initial injury severity we lack the statistical power to fully investigate this question. We have included this analysis in the document pointing out the caveats of it. Future work should address this, as we agree with the reviewers will be important for guiding precision in clinical recommendations.

16) Logistic regression – Based on Figure 2p the overall trend looks more to be that higher MAP = > Pr(δ AIS grade > 0). The exception is only 2 data points on the top end. It is difficult to determine how robust the notion is that there is a 'too high' component to this data. Indeed it seems that a linear model does quite well for this analysis as well. Please see my comment below but this should be addressed as the concept of also having a 'top cutoff' is an extremely important clinical feature here.

We agree with the reviewers that having a ‘top cutoff” is an extremely important clinical feature. Although the linear model performs better than the unconditional model (no predictors), our results show that the polynomial quadratic model or the spline of degree 2 model outperforms the linear model, both in the variance explained and the LOOCV error. We interpret this as indicating that a non-linear association between MAP and the prediction of AIS increase explains the data better than the linear model. We have expanded our explanation in the results to help the reader. Also, please see our answer below regarding the upper limit of the cutoff.

17) Time outside MAP – The authors use an approach that systematically increases their window in both directions to find their optimal range of 76-104. However, what happens if you then hold 76 and only increase on the upper end? Does this rapidly degrade the relationship? If not, again this would suggest that the evidence for a top end cut-off is not as strong. While I understand the authors briefly looked at this (methods) it seems worth exploring further as this is a critical point for clinical management. I do not see a good reason that the time outside the threshold can not at least be plotted to determine this relationship.

We have included further details on the result section about the potential asymmetry of the range. While it is true that we see similar performance of the polynomial quadratic model (symmetric) and the spline model of degree 2 (asymmetric), we did not have strong evidence to justify either model as being 'more correct', so we chose simplicity as spline models will tend to overfit more than a polynomial. For that reason, we used the symmetric window increase in our study of the threshold range. As suggested by the reviewer, we performed the same analysis but holding the lower limit to 76 and increasing the upper limit by 1. This results in the most predictive range to be 76-117. This suggests that our data and methodology is limited to tune the upper level and that it might be between 104 to 117. We have included this analysis in the results and discuss the implication of it.

18) Data availability – The code and analysis should be made available to the reviewers. It is impossible to determine the accuracy of this type of analysis without it.

We are adding the code that reproduces the entire analysis (including figures) as supplementary material, and the data has been uploaded to odc-sci.org, a community-based repository for SCI research data sharing and publication. The data would be available and active upon acceptance with DOIs (10.34945/F5R59J and 10.34945/F5MG68).

19) Discussion – It seems that the authors should discuss the confound of injury severity being linked with worse hemodynamic instability, and also worse neurological recovery. It would be helpful to include some of the suggested analyses to convince the readers that this confound does not explain the results since it is the most likely alternative explanation.

We have performed the suggested analysis when possible and discussed their results. While we agree that there are limitations in our study, we also contend that the findings with the new analyses are important. Although there are caveats to any observational clinical study, this is the largest study to-date on this topic, with a sample size that is 4-fold larger than the last study in this area. The work increases the body of evidence to guide both the clinical conversation and future prospective research.